# MAXTRON: MASK TRANSFORMER WITH TRAJECTORY ATTENTION FOR VIDEO PANOPTIC SEGMENTATION

## ABSTRACT

Video panoptic segmentation requires consistently segmenting (for both 'thing' and 'stuff' classes) and tracking objects in a video over time. In this work, we present MaXTron, a general framework that exploits **Ma**sk **X**Former with **Tr**ajectory Attenti**on** to tackle the task. MaXTron enriches an off-the-shelf mask transformer by leveraging trajectory attention. The deployed mask transformer takes as input a short clip consisting of only a few frames and predicts the clip-level segmentation. To enhance the temporal consistency, MaXTron employs *within-clip* and *cross-clip* tracking modules, efficiently utilizing trajectory attention. Originally designed for video classification, trajectory attention learns to model the temporal correspondences between neighboring frames and aggregates information along the estimated motion paths. However, it is nontrivial to directly extend trajectory attention to the per-pixel dense prediction tasks due to its quadratic dependency on input size. To alleviate the issue, we propose to adapt the trajectory attention for both the dense pixel features and object queries, aiming to improve the short-term and long-term tracking results, respectively. Particularly, in our within-clip tracking module, we propose axial-trajectory attention that effectively computes the trajectory attention for tracking dense pixels sequentially along the height- and width-axes. The axial decomposition significantly reduces the computational complexity for dense pixel features. In our cross-clip tracking module, since the object queries in mask transformer are learned to encode the object information, we are able to capture the long-term temporal connections by applying trajectory attention to object queries, which learns to track each object across different clips. Without bells and whistles, MaXTron demonstrates state-of-the-art performances on video segmentation benchmarks. Code will be publicly available.

## 1 INTRODUCTION

Video panoptic segmentation (Kim et al., 2020) is a challenging computer vision task that requires temporally consistent pixel-level scene understanding by jointly segmenting objects of both 'thing' (*e.g.*, person, car) and 'stuff' classes (*e.g.*, sky, grass), and associating them (*i.e.*, tracking 'thing' objects) across all frames in a video. It can benefit the wide-ranging downstream applications, such as autonomous driving, robot visual control, and video editing. Numerous approaches have been proposed to address the task in a variety of ways. They can be categorized into frame-level segmenters (Kim et al., 2020; Wu et al., 2022c), clip-level segmenters (Athar et al., 2020; Qiao et al., 2021), and video-level segmenters (Wang et al., 2021b; Heo et al., 2022), which process the video either in a frame-by-frame, clip-by-clip, or whole-video manner.

Among them, clip-level segmenters draw our special interest, as it innately captures the local motion within a short period of time (a few frames in the same clip) compared to frame-level segmenters as well as avoids the memory constraints incurred by the video-level segmenters when processing long videos. Specifically, clip-level segmenters first pre-process the video into a set of short clips, each consisting of just a few frames. They then predict clip-level segmentation masks and associate them (*i.e.*, tracking objects across clips) to form the final temporally consistent video-level results. The whole pipeline requires two types of tracking: *within-clip* and *cross-clip* tracking. The within-clip tracking can be implicitly achieved by the clip-level segmenters, while the cross-clip tracking aims to merge the clip-level predictions into video-level results. As a result, existing clip-level segmenters (Li et al., 2023b; Shin et al., 2024), directly extending the modern image segmentation models (Cheng

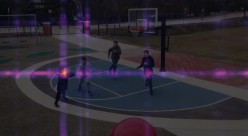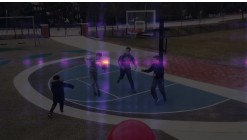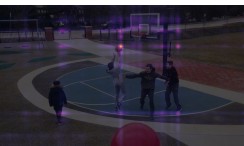

a) *reference point* at frame 1    b) *trajectory attention* at frame 2   c) *trajectory attention* at frame 3   d) *trajectory attention* at frame 4

Figure 1: **Visualization of Learned Axial-Trajectory Attention.** In this short clip of four frames depicting the action 'playing basketball', the basketball location at frame 1 is selected as the *reference point* (mark in red). We compute the axial-trajectory attention sequentially along *H*-axis and *W*-axis. To better understand our method, we multiply the H-axis and W-axis trajectory attentions to visualize the trajectory of the *reference point* over time (*i.e.*, a bright point corresponds to a high attention value in both the H- and W-axis trajectory attention). As shown in the figure, the learned axial-trajectory attention is able to capture the basketball's motion path.

et al., 2022; Yu et al., 2022b) to clip-level segmentation, delegate the duty of within-clip tracking to the clip-level segmenter, and mainly focus on improving the cross-clip tracking to ensure the consistency between neighboring clips. However, they overlook the potential of improving within-clip tracking as well as the long-term consistent tracking beyond neighboring clips.

In this work, we propose MaXTron, a meta-architecture that exploits **Ma**sk **X**Former with **Tr**ajectory Attenti**on** to address the challenges. MaXTron is built on top of an off-the-shelf clip-level segmenter and consists of a *within-clip* tracking module and a *cross-clip* tracking module. The two proposed modules improve the temporal consistency by leveraging trajectory attention (Patrick et al., 2021), which is originally introduced for the task of video classification. Specifically, we adapt the trajectory attention for processing both the dense pixel features and object queries, aiming to improve the within-clip and cross-clip tracking results, respectively.

The trajectory attention learns to model the temporal correspondences between neighboring frames by estimating the motion paths along the time-axis and aggregating information along the trajectories. This property makes trajectory attention a suitable operation to improve the within-clip tracking performance. However, it has computation complexity quadratic to the input size, preventing us from directly applying it to the per-pixel dense video segmentation. Unlike video classification, which pre-processes the input frames into a small set of patch tokens through patch embedding, to tackle this challenge for video segmentation, we propose *axial-trajectory attention* for effectively computing the trajectory attention sequentially along the height- and width-axes in our *within-clip tracking module* and thus unleash the power of trajectory attention for dense pixel-wise tracking (see Fig. 1). Additionally, we enhance our within-clip tracking module by incorporating the multi-scale deformable attention (Zhu et al., 2020), which is stacked iteratively with the proposed axial-trajectory attention to ensure that the learned clip features are both temporally and spatially consistent. Afterwards, the transformer decoder (Cheng et al., 2022; Yu et al., 2022b) is applied to obtain clip-level predictions, where *clip object queries* are learned to encode the objects in each clip (*i.e.*, each query is responsible of predicting an object's mask and semantic class in the clip). At this stage, a clip-level segmenter enhanced with the proposed within-clip tracking module is able to capture motion information within each clip, achieving consistent predictions in a near-online manner.

Moreover, we strengthen the model with a carefully designed *cross-clip tracking module*, which takes all the clip object queries as input. Specifically, each clip is processed by its own set of object queries. Given a video partitioned into several clips, we obtain several sets of object queries from the clip-level segmenter. To capture the whole-video temporal connections, we apply trajectory attention to all the clip object queries. Intuitively, since each object query is learned to encode one object in a clip, applying trajectory attention to all the object queries learns to track each object across different clips through finding its trajectory in the video. In addition to trajectory attention, we also propose the temporal atrous spatial pyramid pooling (Temporal-ASPP) to capture object motion at different time spans. The resulting cross-clip tracking module iteratively stacks trajectory attention and Temporal-ASPP to refine the object queries of a video. It allows us to take the whole video as input during inference, encouraging temporal consistency in a complete offline manner.

In summary, we introduce MaXTron, a simple yet effective unified meta-architecture for video segmentation. MaXTron enriches existing clip-level segmenters by introducing a *within-clip* tracking module and a *cross-clip* tracking module, thus achieving much better temporally consistent segmentation results. We instantiate MaXTron by employing either Video-kMaX (Shin et al., 2024) or

Tube-Link (Li et al., 2023b) as the clip-segmenters. Consequently, MaXTron achieves a significant performance improvement on video panoptic segmentation (Kim et al., 2020) and video instance segmentation (Yang et al., 2019) (where only the 'thing' objects are segmented), respectively. Without bells and whistles, MaXTron improves over Video-kMaX (Shin et al., 2024) by 8.5% and 5.2% VPQ on VIPSeg (Miao et al., 2022) with ResNet50 (He et al., 2016) and ConvNeXt-L (Liu et al., 2022b), respectively. Moreover, it also achieves 3.5% VPQ improvement on VIPSeg compared to the concurrent state-of-the-art model DVIS (Zhang et al., 2023), when using ResNet50. We also show that MaXTron can boost the strong baseline Tube-Link (Li et al., 2023b) on video instance segmentation by 0.9% AP, 4.7% AP$^{long}$, and 6.5% AP on Youtube-VIS-2021 (Yang et al., 2021a), Youtube-VIS-2022 (Yang et al., 2022), and OVIS (Qi et al., 2022) with Swin-L (Liu et al., 2021).

## 2 RELATED WORK

**Attention in Video Transformer** The self-attention mechanism (Vaswani et al., 2017) is widely explored in the modern video transformer design (Bertasius et al., 2021; Arnab et al., 2021; Neimark et al., 2021; Fan et al., 2021; Patrick et al., 2021; Liu et al., 2022a; Wang & Torresani, 2022) with a primary focus on video classification to reason about the temporal information contained in the video. While most works treat time as just another dimension and directly apply global space-time attention, specifically, the divided space-time attention (Bertasius et al., 2021) applies temporal attention and spatial attention separately within each block and achieves superior performance while reducing the computational complexity compared to the standard global space-time attention. Deformable video transformer (Wang & Torresani, 2022) exploits the motion displacements encoded in the video codecs (*e.g.*, MPEG-4) to guide where each query should attend in their deformable space-time attention, thus achieving better performance. Trajectory attention (Patrick et al., 2021) learns to capture the motion path of each query along the time dimension. Our work builds on top of trajectory attention and further extends it from a single label video classification to the dense per-pixel video segmentation by incorporating it with axial-attention (Ho et al., 2019; Huang et al., 2019; Wang et al., 2020) to improve the temporal consistency, while keeping the computational cost manageable. We also apply trajectory attention to object queries for efficiently associating cross-clip predictions.

**Video Panoptic Segmentation** Video panoptic segmentation seeks for holistic video understanding including 'thing' and 'stuff' classes. It requires consistently segmenting them and tracking 'thing' instances, where the latter one also serves as the key challenge for video instance segmentation. Both video panoptic and instance segmentation employ similar tracking modules, and thus we briefly introduce them together. Based on the input manner, they can be roughly categorized into frame-level segmenters (Yang et al., 2019; Kim et al., 2020; Yang et al., 2021b; Ke et al., 2021; Fu et al., 2021; Li et al., 2022; Wu et al., 2022c; Huang et al., 2022; Heo et al., 2023; Liu et al., 2023; Ying et al., 2023; Li et al., 2023a), clip-level segmenters (Athar et al., 2020; Qiao et al., 2021; Hwang et al., 2021; Wu et al., 2022a; Athar et al., 2023; Li et al., 2023b; Shin et al., 2024), and video-level segmenters (Wang et al., 2021b; Lin et al., 2021; Wu et al., 2022b; Heo et al., 2022; Zhang et al., 2023). Specifically, TubeFormer (Kim et al., 2022) tackles multiple video segmentation tasks in a unified manner (Wang et al., 2021a), while TarVIS (Athar et al., 2023) proposes task-independent queries. Tube-Link (Li et al., 2023b) exploits contrastive learning to better align the cross-clip predictions. Video-kMaX (Shin et al., 2024) extends the image segmenter (Yu et al., 2022b) for clip-level video segmentation, and introduces a hierarchical location-aware memory buffer for augmenting cross-clip association. VITA (Heo et al., 2022) exhibits a simple video-level segmenter framework by introducing a set of video queries. DVIS (Zhang et al., 2023) proposes a referring tracker to denoise the frame-level predictions and a temporal refiner to reason about long-term tracking relations. Our work focuses specifically on improving clip-level segmenters, and is thus mostly related to the clip-level panoptic segmeters Video-kMaX (Shin et al., 2024) and Tube-Link (Li et al., 2023b). Building on top of them, MaXTron proposes the within-clip and cross-clip tracking modules for enhancing the temporal consistency within each clip and over the whole video, respectively. Our cross-clip tracking module is also similar to VITA (Heo et al., 2022) and DVIS (Zhang et al., 2023) in the sense that object queries are refined to obtain the final video outputs. However, our model builds on top of clip-level segmenters (instead of frame-level segmenters), and we simply use trajectory attention and the proposed Temporal-ASPP to refine the object queries, while VITA introduces another set of video queries and DVIS additionally cross-attends to the queries cashed in the memory.

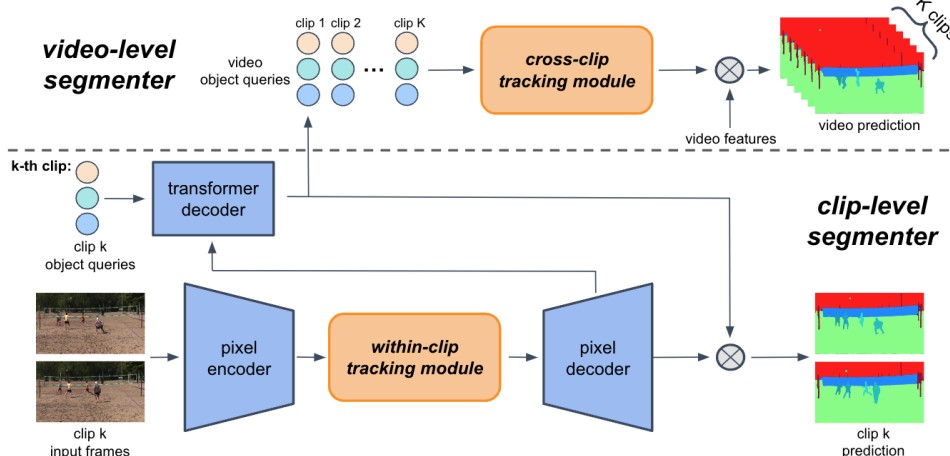

Figure 2: **Overview of MaXTron,** which builds two components on top of a clip-level segmenter (blue): the *within-clip tracking* and *cross-clip tracking modules* (orange). The within-clip tracking module exploits axial-trajectory attention and multi-scale deformable attention for enhancing the local consistency within neighboring frames, while the cross-clip tracking module improves long-term consistency using trajectory attention along with Temporal-ASPP. We obtain video features by concatenating all clip features output by the pixel decoder (totally $K$ clips), and video prediction by multiplying ($\otimes$) video features and refined clip object queries.

## 3 METHOD

In this section, we briefly overview the video segmentation framework that exploits the clip-level segmenter in Sec. 3.1. We then introduce the proposed *within-clip* tracking and *cross-clip* tracking modules for the clip-level segmenter in Sec. 3.2 and Sec. 3.3, respectively.

### 3.1 VIDEO SEGMENTATION WITH CLIP-LEVEL SEGMENTER

**Formulation of Video Segmentation**    Recent works (Kim et al., 2022; Li et al., 2022) have unified different video segmentation tasks as a simple set prediction task (Carion et al., 2020), where the input video is segmented into a set of tubes (a tube is obtained by linking segmentation masks along the time axis) to match the ground-truth tubes. Concretely, given an input video $V \in \mathbb{R}^{L \times 3 \times H \times W}$ with $L$ represents the video length and $H, W$ represent the frame height and width, video segmentation aims at segmenting it into a set of $N$ class-labeled tubes:

$$\{\hat{y}_i\} = \{(\hat{m}_i, \hat{p}_i(c))\}_{i=1}^N, \tag{1}$$

where $\hat{m}_i \in [0, 1]^{L \times H \times W}$ and $\hat{p}_i(c)$ represent the predicted tube and its corresponding semantic class probability. The ground-truth set containing $M$ class-labeled tubes is similarly represented as $\{y_i\} = \{(m_i, p_i(c))\}_{i=1}^M$. These two sets are matched through Hungarian Matching (Kuhn, 1955) during training to compute the losses.

**Formulation of Clip-Level Video Segmentation**    The above video segmentation formulation is theoretically applicable to any length $L$ of video sequences. However, in practice, it is infeasible to fit the whole video into modern large network backbones (Liu et al., 2021) during training. As a result, most works exploit frame-level segmenter (Cheng et al., 2022; Huang et al., 2022) or clip-level segmenter (Qiao et al., 2021; Kim et al., 2022) (a clip is a short video sequence typically of two or three frames) to get frame-level or clip-level tubes first and further associate them to obtain the final video-level tubes. In this work, we focus on the clip-level segmenter, since it better captures local temporal information between frames in the same clip. Formally, we split the whole video $V$ into a set of *non-overlapping* clips: $v_i \in \mathbb{R}^{T \times 3 \times H \times W}$, where $T$ represents the length of each clip in temporal dimension (assuming that $L$ is divisible by $T$ for simplicity; if not, we simply duplicate the last frame). For the clip-level segmenter, we require $T \geq 2$.

**Overview of Proposed MaXTron**    Given the independently predicted clip-level segmentation, we propose MaXTron, a meta-architecture that builds on top of an off-the-shelf clip-level segmenter (*e.g.*, Video-kMaX (Shin et al., 2024) or Tube-Link (Li et al., 2023b)) to generate the final temporally

consistent video-level segmentation results. Building on top of the clip-level segmenter, MaXTron contains two additional modules: within-clip tracking module and cross-clip tracking module, as shown in Fig. 2. We detail each module in the following subsections, and choose Video-kMaX (Shin et al., 2024) as the baseline for simplicity in describing the detailed designs.

## 3.2 WITHIN-CLIP TRACKING MODULE

As shown in Fig. 3, the main component of the within-clip tracking module is the proposed *axial-trajectory attention*, which decomposes the trajectory attention (Patrick et al., 2021) in the height-axis and width-axis, and effectively learns to track objects across the frames in the same clip (thus called *within-clip* tracking). In the module, we also propose to enrich the features by exploiting the multi-scale deformable attention (Zhu et al., 2020). We explain the module in detail below.

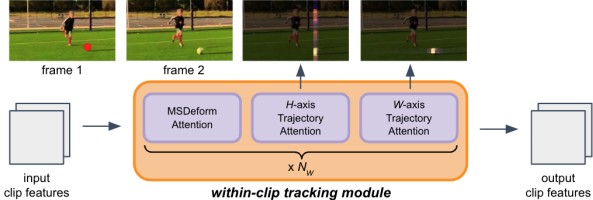

Figure 3: **Within-clip tracking module** iteratively stacks Multi-Scale Deformable Attention and axial-trajectory attention (sequentially along H- and W-axes) for $N_w$ times, and outputs the spatially and temporally consistent clip features. We visualize the attention w.r.t. the reference point (red) at frame 1.

**Axial-Trajectory Attention**  Trajectory attention (Patrick et al., 2021) was originally proposed to capture the object motion information contained in the video for the classification task. However, unlike video classification, where input video is usually pre-processed into a small set of tokens and the output prediction is a single label, video segmentation requires dense prediction (*i.e.*, per pixel) results, making it infeasible to directly apply trajectory attention, which has quadratic complexity proportional to the input size. To unleash the potential of trajectory attention in video segmentation, we propose *axial-trajectory attention* that deploys trajectory attention in a manner similar to axial-attention (Ho et al., 2019; Huang et al., 2019; Wang et al., 2020), which not only effectively captures object motion information but also reduces the computational cost.

Formally, given an input video clip consisting of $T$ frames, we forward it through a frame-level network backbone (*e.g.*, ConvNeXt (Liu et al., 2022b)) to extract the feature map $F \in \mathbb{R}^{T \times D \times H \times W}$, where $D, H, W$ stand for the dimension, height and width of the feature map, respectively. We note that the feature map $F$ is extracted frame-by-frame via the network backbone, and thus no temporal information exchange between frames. We further reshape the feature into $F_h \in \mathbb{R}^{W \times TH \times D}$ to obtain a sequence of $TH$ pixel features $\mathbf{x}_{th} \in \mathbb{R}^D$. Following (Vaswani et al., 2017), we linearly project $\mathbf{x}_{th}$ to a set of query-key-value vectors $\mathbf{q}_{th}, \mathbf{k}_{th}, \mathbf{v}_{th} \in \mathbb{R}^D$. We then perform *axial-attention* along trajectories (*i.e.*, the probabilistic path of a point between frames as defined by (Patrick et al., 2021)). Specifically, for each *reference point* at a specific time-height $th$ position and its corresponding query $\mathbf{q}_{th}$, we construct a set of trajectory points $\widetilde{y}_{tt'h}$ which represents the pooled information weighted by the trajectory probability. The *axial-trajectory* extends for the duration of the video clip, and its point $\widetilde{y}_{tt'h} \in \mathbb{R}^D$ at different times $t'$ is defined as follows:

$$\widetilde{\mathbf{y}}_{tt'h} = \sum_{h'} \mathbf{v}_{t'h'} \cdot \frac{\exp \langle \mathbf{q}_{th}, \mathbf{k}_{t'h'} \rangle}{\sum_{\overline{h}} \exp \langle \mathbf{q}_{th}, \mathbf{k}_{t'\overline{h}} \rangle}. \tag{2}$$

Note that this step computes the axial-trajectory attention in $H$-axis (index $h'$), independently for each frame. It finds the axial-trajectory path of the reference point $th$ across frames $t'$ in the clip by comparing the reference point's query $\mathbf{q}_{th}$ to the keys $\mathbf{k}_{t'h'}$, *only* along the $H$-axis.

To reason about the intra-clip connections, we further pool the trajectories over time $t'$. Specifically, we linearly project the trajectory points $\widetilde{y}_{tt'h}$ and obtain a new set of query-key-value vectors:

$$\widetilde{\mathbf{q}}_{th} = \mathbf{W}_q \widetilde{\mathbf{y}}_{tth}, \quad \widetilde{\mathbf{k}}_{tt'h} = \mathbf{W}_k \widetilde{\mathbf{y}}_{tt'h}, \quad \widetilde{\mathbf{v}}_{tt'h} = \mathbf{W}_v \widetilde{\mathbf{y}}_{tt'h}, \tag{3}$$

where $\mathbf{W}_q, \mathbf{W}_k$, and $\mathbf{W}_v$ are the linear projection matrices for query, key, and value. We then update the reference point at time-height $th$ position by applying 1D attention along the time $t'$:

$$\mathbf{y}_{th} = \sum_{t'} \widetilde{\mathbf{v}}_{tt'h} \cdot \frac{\exp \langle \widetilde{\mathbf{q}}_{th}, \widetilde{\mathbf{k}}_{tt'h} \rangle}{\sum_{\overline{t}} \exp \langle \widetilde{\mathbf{q}}_{th}, \widetilde{\mathbf{k}}_{t\overline{t}h} \rangle}. \tag{4}$$

Figure 4: **Cross-clip tracking module** refines the K sets of clip object queries by iteratively performing trajectory attention and temporal atrous spatial pyramid pooing (Temporal-ASPP) for $N_c$ times.

With the above update rules, we propagate the motion information in *H*-axis in the video clip. To capture global information, we further reshape the feature into $F_w \in \mathbb{R}^{H \times TW \times D}$ and apply the same axial-trajectory attention (but along the $W$-axis) consecutively to capture the width dynamics as well.

The proposed axial-trajectory attention effectively reduces the computational complexity of original trajectory attention from $\mathcal{O}(T^2 H^2 W^2)$ to $\mathcal{O}(T^2 H^2 W + T^2 W^2 H)$, allowing us to apply it to the dense video feature maps, and to reason about the motion information across frames in the same clip.

**Within-Clip Tracking Module** To enhance the features spatially, we further adopt the multi-scale deformable attention (Zhu et al., 2020) for exchanging information at different scales of feature. Specifically, we apply the multi-scale deformable attention to the feature map $F$ (extracted by the network backbone) frame-by-frame, which effectively exchanges the information across feature map scales (stride 32, 16, and 8) for each frame. In the end, the proposed within-clip tracking module is obtained by iteratively stacking multi-scale deformable attention and the proposed axial-trajectory attention (for $N_w$ times) to ensure that the learned features are spatially consistent across the scales and temporally consistent across the frames in the same clip.

**Transformer Decoder** After extracting the spatially and temporally enhanced features, we follow typical video mask transformers (*e.g.*, Video-kMaX (Shin et al., 2024) or Tube-Link (Li et al., 2023b)) to produce clip-level predictions, where clip object queries $C_k \in \mathbb{R}^{N \times D}$ (for $k$-th clip) are iteratively refined by multiple transformer decoder layers (Carion et al., 2020). The resulting *clip object queries* are used to generate a set of $N$ class-labeled tubes within the clip, as described in Sec. 3.1.

**Clip-Level (Near-Online) Inference** With the above within-clip tracking module, our clip-level segmenter is capable of segmenting the video in a near-online fashion (*i.e.*, clip-by-clip). Unlike Video-kMaX (Shin et al., 2024) which takes overlapping clips as input and uses video stitching (Qiao et al., 2021) to link predicted clip-level tubes, our method simply uses the Hungarian Matching (Kuhn, 1955) to associate the clip-level tubes via the clip object queries (similar to MinVIS (Huang et al., 2022); but we work on the clip-level, instead of frame-level), since our input clips are non-overlapping.

### 3.3 Cross-Clip Tracking Module

Though axial-trajectory attention along with the multi-scale deformable attention effectively improves the within-clip tracking ability, the inconsistency between clips (*i.e.*, beyond the clip length $T$) still remains a challenging problem, especially under the fast-moving or occluded scenes. To address these issues, we further propose a cross-clip tracking module to refine and better associate the clip-level predictions. Concretely, given all the clip object queries $\{C_k\}_{k=1}^{K} \in \mathbb{R}^{KN \times D}$ of a video (which is divided into $K = L/T$ non-overlapping clips, and $k$-th clip has its own clip object queries $C_k \in \mathbb{R}^{N \times D}$), we first use the Hungarian Matching to align the clip object queries as the initial tracking results (*i.e.*, "clip-level inference" in Sec. 3.2). Afterwards, they are refined by our proposed cross-clip tracking module to capture whole-video temporal connections (*i.e.*, cross all clips). As shown in Fig. 4, the proposed cross-clip tracking module contains two operations: trajectory-attention and Temporal Atrous Spatial Pyramid Pooling (Temporal-ASPP). We explain each operation below.

**Trajectory Attention** For $k$-th clip, the clip object queries $C_k$ encode the clip-level tube predictions (*i.e.*, each query in $C_k$ generates the class-labeled tube for a certain object in $k$-th clip). Therefore, associating clip-level prediction results is similar to finding the trajectory path of object queries in the whole video. Motivated by this observation, we propose to also exploit trajectory attention (Patrick et al., 2021) for capturing the whole-video temporal connections between clips. Formally, for a video divided into $K$ clips (and each clip is processed by $N$ object queries), each object query $C_{kn} \in \{C_k\}$ is first projected into a set of query-key-value vectors $\mathbf{q}_{kn}, \mathbf{k}_{kn}, \mathbf{v}_{kn} \in \mathbb{R}^D$. Then we compute a set

of trajectory queries $\widetilde{Z}_{kk'n}$ by calculating the probabilistic path of each object query:

$$\widetilde{Z}_{kk'n} = \sum_{k'} \mathbf{v}_{k'n'} \cdot \frac{\exp \langle \mathbf{q}_{kn}, \mathbf{k}_{k'n'} \rangle}{\sum_{\overline{n}} \exp \langle \mathbf{q}_{kn}, \mathbf{k}_{k'\overline{n}} \rangle}. \tag{5}$$

After further projecting the trajectory queries $\widetilde{Z}_{kk'n}$ into $\widetilde{\mathbf{q}}_{kn}, \widetilde{\mathbf{k}}_{kk'n}, \widetilde{\mathbf{v}}_{kk'n}$ similarly to Equ. (3), we aggregate the whole-video cross-clip connections along the trajectory path of object queries through:

$$Z_{kn} = \sum_{k'} \widetilde{\mathbf{v}}_{kk'n} \cdot \frac{\exp \langle \widetilde{\mathbf{q}}_{kn}, \widetilde{\mathbf{k}}_{kk'n} \rangle}{\sum_{\overline{k}} \exp \langle \widetilde{\mathbf{q}}_{kn}, \widetilde{\mathbf{k}}_{k\overline{k}n} \rangle}. \tag{6}$$

**Temporal-ASPP**   While the above trajectory attention reasons about the whole-video temporal connections, it can be further enriched by a short-term tracking module. Motivated by the success of the atrous spatial pyramid pooling (ASPP (Chen et al., 2017)) in capturing spatially multi-scale context information, we extend it to the temporal domain. Specifically, our Temporal-ASPP module contains three parallel temporal atrous convolutions (Chen et al., 2015) with different rates applied to the updated object queries $Z$ for capturing object motion at different time spans.

**Cross-Clip Tracking Module**   The proposed cross-clip tracking module iteratively stacks the trajectory attention and Temporal-ASPP to refine all the clip object queries $\{C_k\}_{k=1}^K$ of a video, obtaining a temporally consistent prediction at the video-level.

**Video-Level (Offline) Inference**   With the proposed within-clip and cross-clip tracking modules, built on top of any clip-level video segmenter, we can now inference the whole video in an offline fashion by exploiting all the refined clip object queries. We first obtain the video features by concatenating all clip features produced by the pixel decoder (totally $K$ clips). The predicted video-level tubes are then generated by multiplying all the clip object queries with the video features (similar to image mask transformers (Wang et al., 2021a; Yu et al., 2022a)). To obtain the predicted classes for the video-level tubes, we exploit another 1D convolution layer (*i.e.*, the "Temporal 1D Conv" in the top-right of Fig. 4) to generate the temporally weighted class predictions, motivated by the fact that the object queries on the trajectory path should have the same class prediction.

## 4   EXPERIMENTAL RESULTS

We evaluate MaXTron based on two different clip-level segmenters on four widely used video segmentation benchmarks to show its generalizability. Specifically, for video panoptic segmentation (VPS), we build MaXTron based on Video-kMaX (Shin et al., 2024) and report performance on VIPSeg (Miao et al., 2022). We also build MaXTron on top of Tube-Link (Li et al., 2023b) for video instance segmentation (VIS) and report the performance on Youtube-VIS 2021 (Yang et al., 2021a), 2022 (Yang et al., 2022), and OVIS (Qi et al., 2022). We follow the original setting of Video-kMaX and Tube-Link to use the same training losses. Note that when training the cross-clip tracking module, both the clip-level segmenter and the within-clip tracking module are frozen due to memory constraint. We provide more implementation details in the appendix.

### 4.1   IMPROVEMENTS OVER BASELINES

We first provide a systematic study to validate the effectiveness of the proposed modules.

**Video Panoptic Segmentation (VPS)**   Tab. 1 summarizes the improvements over the baseline Video-kMaX (Shin et al., 2024) on the VIPSeg dataset. To have a fair comparison, we first reproduce Video-kMaX in our PyTorch framework (which was originally implemented in TensorFlow (Weber et al., 2021a)). Our re-implementation yields significantly better VPQ results, compared to the original model (*e.g.*, 4.5% VPQ improvement with ResNet50), establishing a solid baseline. As shown in the table, using the proposed within-in clip tracking module improves over the reproduced solid baseline by 3.4% and 3.5% VPQ with ResNet50 and ConvNeXt-L, respectively. Employing the proposed cross-clip tracking module further improves the performance by additional 0.6% and 0.9% VPQ with ResNet50 and ConvNeXt-L, respectively. Finally, using the modern ConvNeXtV2-L brings another 1.5% and 0.9% improvements, when compared to the ConvNeXt-L counterparts.

**Video Instance Segmentation (VIS)**   Tab. 2 summarizes the improvements over the baseline Tube-Link (Li et al., 2023b) on the Youtube-VIS-21, -22, and OVIS datasets. Similarly, to have

| method | backbone | RP | WC | CC | VPQ | VPQ$^{Th}$ | VPQ$^{St}$ | VPQ$^1$ | VPQ$^2$ | VPQ$^4$ | VPQ$^6$ |
|---|---|---|---|---|---|---|---|---|---|---|---|
| Video-kMaX | ResNet50 | - | - | - | 38.2 | - | - | - | - | - | - |
| Video-kMaX | ResNet50 | ✓ | - | - | 42.7 | 42.5 | 42.9 | 46.6 | 43.7 | 40.9 | 39.8 |
| MaXTron | ResNet50 | ✓ | ✓ | - | 46.1 | 45.6 | 46.6 | 47.1 | 46.4 | 45.8 | 45.3 |
| MaXTron | ResNet50 | ✓ | ✓ | ✓ | 46.7 | 46.7 | 46.6 | 47.8 | 47.0 | 46.2 | 45.7 |
| Video-kMaX | ConvNeXt-L | - | - | - | 51.9 | - | - | - | - | - | - |
| Video-kMaX | ConvNeXt-L | ✓ | - | - | 52.7 | 54.1 | 51.3 | 55.7 | 53.9 | 51.8 | 49.4 |
| MaXTron | ConvNeXt-L | ✓ | ✓ | - | 56.2 | 58.4 | 54.0 | 56.8 | 56.3 | 55.8 | 55.5 |
| MaXTron | ConvNeXt-L | ✓ | ✓ | ✓ | 57.1 | 59.3 | 54.8 | 57.9 | 57.0 | 56.6 | 56.3 |
| MaXTron | ConvNeXtV2-L | ✓ | ✓ | - | 57.7 | 58.3 | 57.1 | 58.6 | 58.0 | 57.3 | 56.9 |
| MaXTron | ConvNeXtV2-L | ✓ | ✓ | ✓ | 58.0 | 58.8 | 57.2 | 59.0 | 58.3 | 57.6 | 57.0 |

a) **[VPS] VIPSeg val set**

Table 1: **Video Panoptic Segmentation (VPS) results.** We reproduce baseline Video-kMaX (column **RP**) by taking non-overlapping clips as input and replacing their hierarchical matching scheme with simple Hungarian Matching on object queries. **WC**: Our Within-Clip tracking module. **CC**: Our Cross-Clip tracking module.

a fair comparison, we first reproduce the Tube-Link results, using their official code-base. Our reproduction yields similar performances to the original model, except OVIS, where we observe a gap of 4.1% AP for ResNet50. On Youtube-VIS-21 (Tab. 2a), the proposed within-clip tracking module improves the reproduced baselines by 0.6% and 0.6% for ResNet50 and Swin-L, respectively. We note that Tube-Link builds on top of Mask2Former (Cheng et al., 2021), which already adopts six MSDeformAttn layers; thus, it is a even stronger baseline to improve upon. Using our cross-clip tracking module additionally improves the performance by 0.1% and 0.3% for ResNet50 and Swin-L, respectively. On Youtube-VIS-22 (Tab. 2b), our proposed modules bring more significant improvements, showing our method's ability to handle the challenging long videos in the dataset. Specifically, using our within-clip tracking module shows 4.4% and 1.7% AP$^{long}$ for ResNet50 and Swin-L, respectively. Our cross-clip tracking module further improves the performances by 0.5% and 3.0% AP$^{long}$ for ResNet50 and Swin-L, respectively. On OVIS (Tab. 2c), even though we did not successfully reproduce Tube-Link (using their provided config files), we still observe a significant improvement brought by the proposed modules over the reproduced baselines. Particularly, our within-clip tracking modules improves the baselines by 2.2% and 5.8% AP for ResNet50 and Swin-L, respectively. Another improvements of 0.7% and 0.7% AP for ResNet50 and Swin-L can be attained with the proposed cross-clip tracking module. To summarize, our proposed modules bring more remarkable improvements for long and challenging datasets, such as Youtbue-VIS-22 and OVIS.

## 4.2 COMPARISONS WITH OTHER METHODS

After analyzing the improvements brought by the proposed modules, we now move on to compare our MaXTron with other state-of-the-art methods.

**Video Panoptic Segmentation (VPS)** As shown in Tab. 3, in the online/near-online setting, when using ResNet50, our MaXTron significantly outperforms TarVIS (Athar et al., 2023) (which co-trains and exploits multiple video segmentation datasets) by a large margin of 12.6% VPQ. MaXTron also performs better than the very recent ICCV 2023 work DVIS (Zhang et al., 2023) (trained with 5 frames and tested with 1 frame) by a healthy margin of 6.9% VPQ. When using the stronger backbones, MaXTron with ConvNeXt-L still outperforms TarVIS and DVIS with Swin-L by 8.2% and 1.5% VPQ, respectively. The performance is further improved by using the modern ConvNeXtV2-L backbone, attaining 57.7% VPQ. In the offline setting, MaXTron with ResNet50 outperforms DVIS by 3.5% VPQ, while MaXTron with ConvNeXt-L performs comparably to DVIS with Swin-L. Finally, when using the modern ConvNeXtV2-L, MaXTron achieves 58.0% VPQ, setting a new state-of-the-art.

**Video Instance Segmentation (VIS)** In Tab. 4, we compare MaXTron with other state-of-the-art methods for VIS. On Youtube-VIS-21 (Tab. 4a), MaXTron slightly outperforms TarVIS (Athar et al., 2023) and DVIS (Zhang et al., 2023) by 0.1% and 1.1% AP, respectively. On Youtube-VIS-22 (Tab. 4b), MaXTron performs better than DVIS in both online/near-online and offline settings by 5.3% and 1.1% AP$^{long}$, respectively.

| method | backbone | RP | WC | CC | AP |
|---|---|---|---|---|---|
| Tube-Link | ResNet50 | - | - | - | 47.9 |
| Tube-Link | ResNet50 | ✓ | - | - | 47.8 |
| MaXTron | ResNet50 | ✓ | ✓ | - | 48.4 |
| MaXTron | ResNet50 | ✓ | ✓ | ✓ | 48.5 |
| Tube-Link | Swin-L | - | - | - | 58.4 |
| Tube-Link | Swin-L | ✓ | - | - | 58.2 |
| MaXTron | Swin-L | ✓ | ✓ | - | 58.8 |
| MaXTron | Swin-L | ✓ | ✓ | ✓ | 59.1 |

a) **[VIS] Youtube-VIS-21 val set**

| method | backbone | RP | WC | CC | $AP^{long}$ |
|---|---|---|---|---|---|
| Tube-Link | ResNet50 | - | - | - | 31.1 |
| Tube-Link | ResNet50 | ✓ | - | - | 32.1 |
| MaXTron | ResNet50 | ✓ | ✓ | - | 36.5 |
| MaXTron | ResNet50 | ✓ | ✓ | ✓ | 37.0 |
| Tube-Link | Swin-L | - | - | - | 34.2 |
| Tube-Link | Swin-L | ✓ | - | - | 34.2 |
| MaXTron | Swin-L | ✓ | ✓ | - | 35.9 |
| MaXTron | Swin-L | ✓ | ✓ | ✓ | 38.9 |

b) **[VIS] Youtube-VIS-22 val set**

| method | backbone | RP | WC | CC | AP |
|---|---|---|---|---|---|
| Tube-Link | ResNet50 | - | - | - | 29.5 |
| Tube-Link | ResNet50 | ✓ | - | - | 25.4§ |
| MaXTron | ResNet50 | ✓ | ✓ | - | 27.6 |
| MaXTron | ResNet50 | ✓ | ✓ | ✓ | 28.3 |
| Tube-Link | Swin-L | - | - | - | N/A |
| Tube-Link | Swin-L | ✓ | - | - | 33.3 |
| MaXTron | Swin-L | ✓ | ✓ | - | 39.1 |
| MaXTron | Swin-L | ✓ | ✓ | ✓ | 39.8 |

c) **[VIS] OVIS val set**

Table 2: **Video Instance Segmentation (VIS) results.** We reproduce baseline Tube-Link (column **RP**) with their official code-base. We then build on top of it with our Within-Clip tracking module (**WC**) and Cross-Clip tracking module (**CC**). For Youtube-VIS-22, we mainly report $AP^{long}$ (for long videos) and see appendix for $AP^{short}$ (for short videos) and $AP^{all}$ (average of them). §: Our best attempt to reproduce Tube-Link's performances (25.4%), lower than the results (29.5%) reported in the paper. Their provided checkpoints also yield lower results (26.7%). N/A: Not available from their code-base, but we have attempted to reproduce.

| method | backbone | VPQ | $VPQ^{Th}$ | $VPQ^{St}$ |
|---|---|---|---|---|
| *online/near-online methods* | | | | |
| TarVIS (Athar et al., 2023) | ResNet50 | 33.5 | 39.2 | 28.5 |
| DVIS (Zhang et al., 2023)†‡ | ResNet50 | 39.2 | 39.3 | 39.0 |
| TarVIS (Athar et al., 2023) | Swin-L | 48.0 | 58.2 | 39.0 |
| DVIS (Zhang et al., 2023)† | Swin-L | 54.7 | 54.8 | 54.6 |
| MaXTron w/ Video-kMaX | ResNet50 | 46.1 | 45.6 | 46.6 |
| MaXTron w/ Video-kMaX | ConvNeXt-L | 56.2 | 58.4 | 54.0 |
| MaXTron w/ Video-kMaX | ConvNeXtV2-L | 57.7 | 58.3 | 57.1 |
| *offline methods* | | | | |
| DVIS (Zhang et al., 2023)† | ResNet50 | 43.2 | 43.6 | 42.8 |
| DVIS (Zhang et al., 2023)† | Swin-L | 57.6 | 59.9 | 55.5 |
| MaXTron w/ Video-kMaX | ResNet50 | 46.7 | 46.7 | 46.6 |
| MaXTron w/ Video-kMaX | ConvNeXt-L | 57.1 | 59.3 | 54.8 |
| MaXTron w/ Video-kMaX | ConvNeXtV2-L | 58.0 | 58.8 | 57.2 |

a) **[VPS] VIPSeg val set**

Table 3: **Video Panoptic Segmentation (VPS) results.** We compare our MaXTron with other state-of-the-art works. †: Very recent ICCV 2023 work. ‡: Evaluated using their open-source checkpoint.

| method | backbone | AP |
|---|---|---|
| *online/near-online methods* | | |
| TarVIS (Athar et al., 2023) | ResNet50 | 48.3 |
| MaXTron | ResNet50 | 48.4 |
| *offline methods* | | |
| VITA (Heo et al., 2022) | ResNet50 | 45.7 |
| DVIS (Zhang et al., 2023)† | ResNet50 | 47.4 |
| MaXTron | ResNet50 | 48.5 |

a) **[VIS] Youtube-VIS-21 val set**

| method | backbone | $AP^{long}$ |
|---|---|---|
| *online/near-online methods* | | |
| DVIS (Zhang et al., 2023)†* | ResNet50 | 31.2 |
| MaXTron | ResNet50 | 36.5 |
| *offline methods* | | |
| VITA (Heo et al., 2022)* | ResNet50 | 31.9 |
| DVIS (Zhang et al., 2023)†* | ResNet50 | 35.9 |
| MaXTron | ResNet50 | 37.0 |

b) **[VIS] Youtube-VIS-22 val set**

Table 4: **Video Instance Segmentation (VIS) results.** We compare our MaXTron with other state-of-the-art works. †: Very recent ICCV 2023 work. *: All results are reproduced by us using their official checkpoints.

## 5 CONCLUSION

We have presented MaXTron, a meta-architecture that enhances an off-the-shelf clip-level segmenter with the proposed within-clip and cross-clip tracking modules, which encourage short-term and long-term temporal consistency by leveraging trajectory attention. Specifically, the within-clip tracking module employs the proposed axial-trajectory attention, which efficiently computes the trajectory attention sequentially along the height- and width-axes, while the cross-clip tracking module exploits the trajectory attention along with the proposed Temporal-ASPP to refine object queries. Consequently, MaXTron demonstrates state-of-the-art performances on the video segmentation benchmarks. We hope our work can inspire more research on efficient attentions for video segmentation.

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

APPENDIX

In the appendix, we provide additional information as listed below:

- Sec. A provides the dataset information.
- Sec. B provides the implementation details.
- Sec. C provides additional experimental results, including ablation studies and more comparison with other methods for video panoptic segmentation (VPS) and video instance segmentation (VIS).
- Sec. D provides visualization results.
- Sec. E discusses our method's limitations.
- Sec. F provides additional materials to address the concerns in reviews.

## A    DATASETS

In this section, we provide more details of the experimented datasets.

**VIPSeg** (Miao et al., 2022) is a new large-scale video panoptic segmentation dataset, targeting for diverse in-the-wild scenes. The dataset contains 124 semantic classes (58 'thing' and 66 'stuff' classes) with 3536 videos, where each video spans 3 to 10 seconds. The main adopted evaluation metric is VPQ (video panoptic quality) (Kim et al., 2020) on this benchmark.

License: The data is released for non-commercial research purpose only.

URL: https://github.com/VIPSeg-Dataset/VIPSeg-Dataset

**Youtube-VIS** (Yang et al., 2019) is a popular benchmark on video instance segmentation (where only 'thing' classes are segmented and tracked). It contains multiple versions. The YouTube-VIS-2019 (Yang et al., 2019) consists of 40 semantic classes, while the YouTube-VIS-2021 (Yang et al., 2021a) and YouTube-VIS-2022 (Yang et al., 2022) are improved versions with higher number of instances and videos. Youtube-VIS adopts track AP (Yang et al., 2019) for evaluation.

License: Creative Commons Attribution 4.0

URL: https://youtube-vos.org/dataset/vis/

**OVIS** (Qi et al., 2022) is a challenging video instance segmentation dataset with focuses on long videos (12.77 seconds on average), and objects with severe occlusion and complex motion patterns. The dataset contains 25 semantic classes and also adopt track AP (Yang et al., 2019) for evaluation.

License: CC BY-NC-SA 4.0

URL: https://songbai.site/ovis/

## B    IMPLEMENTATION DETAILS

**Implementation Details**    The proposed MaXTron is a unified approach for both near-online and offline video segmentation (*i.e.*, the cross-clip tracking module is only used for the offline setting). For the near-online setting (*i.e.*, employing the within-clip tracking module), we use a clip size of two and four for VPS and VIS, respectively. For the offline setting (*i.e.*, employing the cross-clip tracking module), we adopt a video length of 24 (*i.e.*, 12 clips) for VPS and 20 (*i.e.*, 5 clips) for VIS. At this stage, we only train the cross-clip tracking module, while both the clip-level segmenter and the within-clip tracking module are frozen due to memory constraint. During testing, we directly inference with the whole video with our full model.

We experiment with four backbones for MaXTron: ResNet50 (He et al., 2016), Swin-L (Liu et al., 2021), ConvNeXt-L (Liu et al., 2022b) and ConvNeXt V2-L (Woo et al., 2023). For VPS experiments, we first reproduce Video-kMaX (Shin et al., 2024) based on the official PyTorch re-implementation of kMaX-DeepLab (Yu et al., 2022b). We employ a specific pre-training protocol for VIPSeg, closely following the prior works (Weber et al., 2021b; Kim et al., 2022; Shin et al., 2024). Concretely,

starting with an ImageNet (Russakovsky et al., 2015) pre-trained backbone, we pre-train the kMaX-DeepLab and Multi-Scale Deformable Attention (MSDeformAttn) in our within-clip tracking module on COCO (Lin et al., 2014). The within-clip and cross-clip tracking modules deploy $N_w = 2$ and $N_c = 6$ blocks, respectively, for VPS. On the other hand, for VIS experiments, we use the official code-base of Tube-Link (Li et al., 2023b). Since Tube-Link is built on top of Mask2Former (Cheng et al., 2022) and thus already contains six layers of MSDeformAttn, we simplify our within-clip tracking module by directly inserting axial-trajectory attention after each original MSDeformAttn. As a result, the within-clip and cross-clip tracking modules use $N_w = 6$ and $N_c = 4$ blocks, respectively, for VIS. We note that we do not use any other video datasets (*e.g.*, pseudo COCO videos) for pre-training axial-trajectory attention.

## C  ADDITIONAL EXPERIMENTAL RESULTS

In this section, we provide more experimental results, including the ablation studies on the proposed within-clip and cross-clip tracking modules (Sec. C.1), as well as more detailed comparisons with other state-of-the-art methods (Sec. C.2).

### C.1  ABLATION STUDIES

We conduct the ablations studies on the VIPSeg dataset due to its scene diversity and long-length videos, using ResNet50 (He et al., 2016).

**Within-Clip Tracking Module**  In Tab. 5, we ablate the design choices of the proposed within-clip tracking module. To begin with, we employ one MSDeformAttn (Multi-Scale Deformable Attention) and one TrjAttn (Trajectory Attention) with $N_w = 2$ (*i.e.*, stacking two blocks of them), obtaining the performance of 45.3% VPQ. Replacing the TrjAttn with the proposed AxialTrjAttn (Axial Trajectory Attention, sequentially along $H$- and $W$-axes) yields a comparable performance of 45.4%. Stacking two AxialTrjAttn layers in each block leads to our final setting with 46.1%. We note that it will be Out-Of-Memory, if we stack two TrjAttn layers in a V100 GPU. Increasing or decreasing the number of blocks $N_w$ degrades the performance slightly. If we employ one more AxialTrjAttn layers per block, the performance drops by 0.4%. Finally, if we change the iterative stacking scheme to a sequential manner (*i.e.*, stacking two MSDeformAttn, followed by four AxialTrjAttn), the performance also decreases slightly by 0.3%.

**Cross-Clip Tracking Module**  Tab. 6 summarizes our ablation studies on the design choices of the proposed cross-clip tracking module. Particularly, in Tab. 6a, we adopt different operations in the module. Using self-attention (Self-Attn), instead of trajectory attention (TrjAttn) degrades the performance by 0.3% VPQ. Removing the Temporal-ASPP operation also decreases the performance by 0.2%. In Tab. 6b, we ablate the atrous rates used in the three parallel temporal convolutions (with kernel size 3) of the proposed Temporal-ASPP. Using atrous rates $(1, 2, 3)$ (*i.e.*, rates set to 1, 2, and 3 for those three convolutions, respectively) leads to the best performance. In Tab. 6c, we find that using $N_c = 6$ blocks in the cross-clip tracking module yields the best result.

### C.2  COMPARISONS WITH OTHER METHODS

**Video Panoptic Segmentation (VPS)**  In Tab. 7, we compare with more state-of-the-art methods on the VIPSeg dataset. We observe the similar trend as discussed in the main paper, and thus simply list all the other methods for a complete comparison.

**Video Instance Segmentation (VIS)**  In Tab. 8, we report more state-of-the-art methods on the Youtube-VIS-21 dataset. As shown in the table, our MaXTron with ResNet50 backbone demonstrates a better performance than the other methods (as discussed in the main paper), while our MaXTron with Swin-L performs slightly worse than TarVIS (Athar et al., 2023) in the online/near-online setting and than DVIS (Zhang et al., 2023) in the offline setting. We think the performance can be improved by exploiting more video segmentation datasets, as TarVIS did, or by improving the clip-level segmenter (particularly, our baseline Tube-Link with Swin-L performs worse than the other state-of-the-art methods with Swin-L).

For the Youtube-VIS-22 results, we notice that the reported numbers in some recent papers are not comparable, since some papers report $AP^{long}$ (AP for long videos) while some papers use $AP^{all}$, which

| #MSDeformAttn | #TrjAttn | #AxialTrjAttn | $N_w$ | VPQ |
|---|---|---|---|---|
| - | - | - | - | 42.7 |
| 1 | 1 | - | 2 | 45.3 |
| 1 | - | 1 | 2 | 45.4 |
| 1 | - | 2 | 2 | 46.1 |
| 1 | - | 2 | 1 | 44.7 |
| 1 | - | 2 | 3 | 45.2 |
| 1 | - | 3 | 2 | 45.7 |
| 2 | - | 4 | 1 | 45.8 |

Table 5: **Ablation on within-clip tracking module.** We vary the number of Multi-Scale Deformable Attnetion (#MSDeformAttn), number of Trajectory Attention (#TrjAttn), or number of Axial-Trajectory Attention (#Axial-TrjAttn). $N_w$ detnotes the number of blocks (*i.e.*, repetitions). −: Not using any operations. The final setting is marked in grey.

| SelfAttn | TrjAttn | Temporal-ASPP | VPQ |
|---|---|---|---|
| ✓ | | ✓ | 46.4 |
| | ✓ | ✓ | 46.7 |
| | | ✓ | 46.5 |

a) **Operations**

| atrous rates | VPQ |
|---|---|
| (1, 2, 3) | 46.7 |
| (1, 2, 5) | 46.5 |
| (1, 3, 5) | 46.4 |

b) **Temporal-ASPP**

| $N_c$ | VPQ |
|---|---|
| 4 | 46.7 |
| 6 | 46.7 |
| 8 | 46.2 |

c) **Number of blocks** $N_c$

Table 6: **Ablation on cross-clip tracking module.** We vary operations in the block, Temporal-ASPP (atrous rates), and number of blocks $N_c$ (*i.e.*, repetitions). The final setting is marked in grey.

is the average of AP$^{\text{long}}$ and AP$^{\text{short}}$ (AP for short videos). To carefully and fairly compare between methods, we therefore reproduce all the state-of-the-art results by using their official open-source checkpoints, and clearly report their AP$^{\text{all}}$, AP$^{\text{long}}$, and AP$^{\text{short}}$ in Tab. 9. Similar to the discussion in the main paper, our MaXTron with ResNet50 significantly improves over the baseline Tube-Link and performs better than other state-of-the-art methods (particularly in AP$^{\text{long}}$). However, our results with Swin-L lag behind other state-of-the-art methods with Swin-L, whose gap may be bridged by improving the baseline Tube-Link Swin-L.

In Tab. 10, we summarize more comparisons with other state-of-the-art methods on OVIS. As shown in the table, our method remarkably improves over the baseline, but performs worse than the state-of-the-art methods, partially because we fail to fully reproduce the baseline Tube-Link that our method heavily depends upon. Similar to our other VIS results, we think the improvement of clip-level segmenter will also lead to the improvement of MaXTron.

## D VISUALIZATION RESULTS

We provide visualization results in Fig. 5, Fig. 6, Fig. 7, and Fig. 8 for different video sequences. We compare with DVIS (Zhang et al., 2023) and our re-implemented Video-kMaX (Shin et al., 2024) with ResNet50 as backbone and inference the video in an online/near-online fashion.

## E LIMITATIONS

The proposed MaXTron builds on top of off-the-shelf clip-level segmenters with the proposed within-clip and cross-clip tracking modules. Even though flexible, its performance depends on the underlying employed clip-level segmenter. As a result, we foresee any new breakthrough developed for the clip-level segmenter will also in turn improve our method's performances. Additionally, when training the proposed cross-clip tracking module, the clip-level segmenter and the within-clip tracking module are frozen (due to the GPU memory limit), which may lead to a sub-optimal result (ideally, end-to-end training leads to a better performance). We leave it as a future work to efficiently fine-tune the whole model for processing long videos.

| method | backbone | VPQ | VPQ$^{Th}$ | VPQ$^{St}$ |
|---|---|---|---|---|
| *online/near-online methods* | | | | |
| ViP-DeepLab (Qiao et al., 2021) | ResNet50 | 16.0 | - | - |
| VPSNet-FuseTrack (Kim et al., 2020) | ResNet50 | 17.0 | - | - |
| VPSNet-SiamTrack (Woo et al., 2021) | ResNet50 | 17.2 | - | - |
| Clip-PanoFCN (Miao et al., 2022) | ResNet50 | 22.9 | - | - |
| Video K-Net (Li et al., 2022) | ResNet50 | 26.1 | - | - |
| TubeFormer (Kim et al., 2022) | Axial-ResNet50-B3 | 31.2 | - | - |
| TarVIS (Athar et al., 2023) | ResNet50 | 33.5 | 39.2 | 28.5 |
| Video-kMaX (Shin et al., 2024) | ResNet50 | 38.2 | - | - |
| Tube-Link (Li et al., 2023b) | ResNet50 | 39.2 | - | - |
| DVIS (Zhang et al., 2023)†‡ | ResNet50 | 39.2 | 39.3 | 39.0 |
| TarVIS (Athar et al., 2023) | Swin-L | 48.0 | 58.2 | 39.0 |
| Video-kMaX (Shin et al., 2024) | ConvNeXt-L | 51.9 | - | - |
| DVIS (Zhang et al., 2023)† | Swin-L | 54.7 | 54.8 | 54.6 |
| MaXTron w/ Video-kMaX (ours) | ResNet50 | 46.1 | 45.6 | 46.6 |
| MaXTron w/ Video-kMaX (ours) | ConvNeXt-L | 56.2 | 58.4 | 54.0 |
| MaXTron w/ Video-kMaX (ours) | ConvNeXt V2-L | 57.7 | 58.3 | 57.1 |
| *offline methods* | | | | |
| DVIS (Zhang et al., 2023)† | ResNet50 | 43.2 | 43.6 | 42.8 |
| DVIS (Zhang et al., 2023)† | Swin-L | 57.6 | 59.9 | 55.5 |
| MaXTron w/ Video-kMaX (ours) | ResNet50 | 46.7 | 46.7 | 46.6 |
| MaXTron w/ Video-kMaX (ours) | ConvNeXt-L | 57.1 | 59.3 | 54.8 |
| MaXTron w/ Video-kMaX (ours) | ConvNeXt V2-L | 58.0 | 58.8 | 57.2 |

Table 7: **VIPSeg *val* set results.** We provide more complete comparisons with other state-of-the-art methods. †: Very recent ICCV 2023 work. ‡: Evaluated using their open-source checkpoint.

| method | backbone | AP | AP$_{50}$ | AP$_{75}$ | AR$_1$ | AR$_{10}$ |
|---|---|---|---|---|---|---|
| *online/near-online methods* | | | | | | |
| MinVIS (Huang et al., 2022) | ResNet50 | 44.2 | 66.0 | 48.1 | 39.2 | 51.7 |
| IDOL (Wu et al., 2022c) | ResNet50 | 43.9 | 68.0 | 49.6 | 38.0 | 50.9 |
| GenVIS$_{near-online}$ (Heo et al., 2023) | ResNet50 | 46.3 | 67.0 | 50.2 | 40.6 | 53.2 |
| DVIS (Zhang et al., 2023)† | ResNet50 | 46.4 | 68.4 | 49.6 | 39.7 | 53.5 |
| GenVIS$_{online}$ (Heo et al., 2023) | ResNet50 | 47.1 | 67.5 | 51.5 | 41.6 | 54.7 |
| Tube-Link (Li et al., 2023b) | ResNet50 | 47.9 | 70.0 | 50.2 | 42.3 | 55.2 |
| TarVIS (Athar et al., 2023) | ResNet50 | 48.3 | 69.6 | 53.2 | 40.5 | 55.9 |
| MinVIS (Huang et al., 2022) | Swin-L | 55.3 | 76.6 | 62.0 | 45.9 | 60.8 |
| IDOL (Wu et al., 2022c) | Swin-L | 56.1 | 80.8 | 63.5 | 45.0 | 60.1 |
| Tube-Link (Li et al., 2023b) | Swin-L | 58.4 | 79.4 | 64.3 | 47.5 | 63.6 |
| DVIS (Zhang et al., 2023)† | Swin-L | 58.7 | 80.4 | 66.6 | 47.5 | 64.6 |
| GenVIS$_{online}$ (Heo et al., 2023) | Swin-L | 59.6 | 80.9 | 65.8 | 48.7 | 65.0 |
| GenVIS$_{near-online}$ (Heo et al., 2023) | Swin-L | 60.1 | 80.9 | 66.5 | 49.1 | 64.7 |
| TarVIS (Athar et al., 2023) | Swin-L | 60.2 | 81.4 | 67.6 | 47.6 | 64.8 |
| MaXTron w/ Tube-Link (ours) | ResNet50 | 48.4 | 71.1 | 51.8 | 42.0 | 57.4 |
| MaXTron w/ Tube-Link (ours) | Swin-L | 58.8 | 81.3 | 65.0 | 46.7 | 62.7 |
| *offline methods* | | | | | | |
| VITA (Heo et al., 2022) | ResNet50 | 45.7 | 67.4 | 49.5 | 40.9 | 53.6 |
| DVIS (Zhang et al., 2023)† | ResNet50 | 47.4 | 71.0 | 51.6 | 39.9 | 55.2 |
| VITA (Heo et al., 2022) | Swin-L | 57.5 | 80.6 | 61.0 | 47.7 | 62.6 |
| DVIS (Zhang et al., 2023)† | Swin-L | 60.1 | 83.0 | 68.4 | 47.7 | 65.7 |
| MaXTron w/ Tube-Link (ours) | ResNet50 | 48.5 | 70.9 | 52.4 | 42.3 | 57.9 |
| MaXTron w/ Tube-Link (ours) | Swin-L | 59.1 | 81.9 | 64.9 | 46.9 | 63.8 |

Table 8: **Youtube-VIS-21 *val* set results.** We provide more complete comparisons with other state-of-the-art methods. †: Very recent ICCV 2023 work.

| method | backbone | $AP^{all}$ | $AP^{short}$ | $AP_{50}$ | $AP_{75}$ | $AR_1$ | $AR_{10}$ | $AP^{long}$ | $AP_{50}$ | $AP_{75}$ | $AR_1$ | $AR_{10}$ |
|---|---|---|---|---|---|---|---|---|---|---|---|---|
| *online/near-online methods* | | | | | | | | | | | | |
| MinVIS (Huang et al., 2022)[*] | ResNet50 | 32.8 | 43.9 | 66.9 | 47.5 | 38.8 | 51.9 | 21.6 | 42.9 | 18.1 | 18.8 | 25.6 |
| GenVIS$_{near-online}$ (Heo et al., 2023)[*] | ResNet50 | 38.1 | 45.9 | 66.3 | 50.2 | 40.8 | 53.7 | 30.3 | 50.9 | 32.7 | 25.5 | 36.2 |
| DVIS (Zhang et al., 2023)†[*] | ResNet50 | 38.6 | 46.0 | 68.1 | 50.4 | 39.7 | 53.5 | 31.2 | 50.4 | 36.8 | 30.2 | 35.7 |
| Tube-Link (Li et al., 2023b)[*] | ResNet50 | 39.5 | 47.9 | 70.4 | 50.5 | 42.6 | 55.9 | 31.1 | 56.1 | 31.2 | 29.1 | 36.3 |
| MinVIS (Huang et al., 2022)[*] | Swin-L | 43.5 | 55.0 | 77.8 | 60.6 | 45.3 | 60.3 | 31.9 | 51.4 | 33.0 | 28.2 | 35.3 |
| Tube-Link (Li et al., 2023b)[*] | Swin-L | 46.0 | 57.8 | 78.7 | 63.4 | 47.0 | 62.7 | 34.2 | 53.2 | 37.9 | 31.5 | 38.9 |
| DVIS (Zhang et al., 2023)†[*] | Swin-L | 48.9 | 58.8 | 80.6 | 65.9 | 47.5 | 63.9 | 39.0 | 56.0 | 43.0 | 33.0 | 43.5 |
| MaXTron w/ Tube-Link (ours) | ResNet50 | 41.6 | 46.8 | 68.1 | 50.5 | 41.5 | 56.2 | 36.5 | 61.1 | 41.7 | 32.3 | 42.3 |
| MaXTron w/ Tube-Link (ours) | Swin-L | 47.3 | 58.7 | 81.1 | 64.9 | 46.9 | 62.7 | 35.9 | 62.0 | 37.0 | 34.2 | 39.7 |
| *offline methods* | | | | | | | | | | | | |
| VITA (Heo et al., 2022)[*] | ResNet50 | 38.8 | 45.7 | 66.6 | 50.1 | 41.0 | 53.1 | 31.9 | 53.8 | 37.0 | 31.1 | 37.3 |
| DVIS (Zhang et al., 2023)†[*] | ResNet50 | 41.6 | 47.2 | 70.8 | 51.0 | 40.0 | 54.9 | 35.9 | 58.4 | 39.9 | 32.2 | 41.9 |
| VITA (Heo et al., 2022)[*] | Swin-L | 49.3 | 57.6 | 80.4 | 62.5 | 47.7 | 62.3 | 41.0 | 62.1 | 43.9 | 39.4 | 43.5 |
| DVIS (Zhang et al., 2023)†[*] | Swin-L | 52.4 | 59.9 | 82.7 | 68.3 | 47.8 | 65.2 | 44.9 | 66.3 | 48.9 | 37.1 | 53.2 |
| MaXTron w/ Tube-Link (ours) | ResNet50 | 41.3 | 45.6 | 68.0 | 51.1 | 40.2 | 54.7 | 37.0 | 63.4 | 36.7 | 29.0 | 40.2 |
| MaXTron w/ Tube-Link (ours) | Swin-L | 48.8 | 58.7 | 81.0 | 64.2 | 46.6 | 63.5 | 38.9 | 64.4 | 39.3 | 32.0 | 42.3 |

Table 9: **Youtube-VIS-22 *val* set results.** We provide more complete comparisons with other state-of-the-art methods. †: Very recent ICCV 2023 work. [*]: All results are reproduced by us using their official checkpoints. We report $AP^{short}$ and $AP^{long}$ for short and long videos, respectively, and $AP^{all}$ by averaging them.

| method | backbone | AP | $AP_{50}$ | $AP_{75}$ | $AR_1$ | $AR_{10}$ |
|---|---|---|---|---|---|---|
| *online/near-online methods* | | | | | | |
| MinVIS (Huang et al., 2022) | ResNet50 | 25.0 | 45.5 | 24.0 | 13.9 | 29.7 |
| Tube-Link (Li et al., 2023b)[§] | ResNet50 | 25.4 | 44.9 | 26.5 | 14.1 | 30.1 |
| Tube-Link (Li et al., 2023b) | ResNet50 | 29.5 | 51.5 | 30.2 | 15.5 | 34.5 |
| IDOL (Wu et al., 2022c) | ResNet50 | 30.2 | 51.3 | 30.0 | 15.0 | 37.5 |
| DVIS (Zhang et al., 2023)† | ResNet50 | 30.2 | 55.0 | 30.5 | 14.5 | 37.3 |
| TarVIS (Athar et al., 2023) | ResNet50 | 31.1 | 52.5 | 30.4 | 15.9 | 39.9 |
| GenVIS$_{near-online}$ (Heo et al., 2023) | ResNet50 | 34.5 | 59.4 | 35.0 | 16.6 | 38.3 |
| GenVIS$_{online}$ (Heo et al., 2023) | ResNet50 | 35.8 | 60.8 | 36.2 | 16.3 | 39.6 |
| Tube-Link (Li et al., 2023b)[§] | Swin-L | 33.3 | 54.6 | 32.8 | 16.8 | 37.7 |
| MinVIS (Huang et al., 2022) | Swin-L | 39.4 | 61.5 | 41.3 | 18.1 | 43.3 |
| IDOL (Wu et al., 2022c) | Swin-L | 42.6 | 65.7 | 45.2 | 17.9 | 49.6 |
| TarVIS (Athar et al., 2023) | Swin-L | 43.2 | 67.8 | 44.6 | 18.0 | 50.4 |
| GenVIS$_{online}$ (Heo et al., 2023) | Swin-L | 45.2 | 69.1 | 48.4 | 19.1 | 48.6 |
| GenVIS$_{near-online}$ (Heo et al., 2023) | Swin-L | 45.4 | 69.2 | 47.8 | 18.9 | 49.0 |
| DVIS (Zhang et al., 2023)† | Swin-L | 47.1 | 71.9 | 49.2 | 19.4 | 52.5 |
| MaXTron w/ Tube-Link | ResNet50 | 27.6 | 50.1 | 27.2 | 14.6 | 32.5 |
| MaXTron w/ Tube-Link | Swin-L | 39.1 | 62.3 | 39.8 | 18.5 | 42.3 |
| *offline methods* | | | | | | |
| VITA (Heo et al., 2022) | ResNet50 | 19.6 | 41.2 | 17.4 | 11.7 | 26.0 |
| DVIS (Zhang et al., 2023)† | ResNet50 | 33.8 | 60.4 | 33.5 | 15.3 | 39.5 |
| VITA (Heo et al., 2022) | Swin-L | 27.7 | 51.9 | 24.9 | 14.9 | 33.0 |
| DVIS (Zhang et al., 2023)† | Swin-L | 48.6 | 74.7 | 50.5 | 18.8 | 53.8 |
| MaXTron w/ Tube-Link | ResNet50 | 28.3 | 50.7 | 27.0 | 14.6 | 34.0 |
| MaXTron w/ Tube-Link | Swin-L | 39.8 | 64.5 | 40.1 | 17.9 | 43.7 |

Table 10: **OVIS *val* set results.** We provide more complete comparisons with other state-of-the-art methods. [§]: Reproduced by us using their official code-base. †: Very recent ICCV 2023 work.

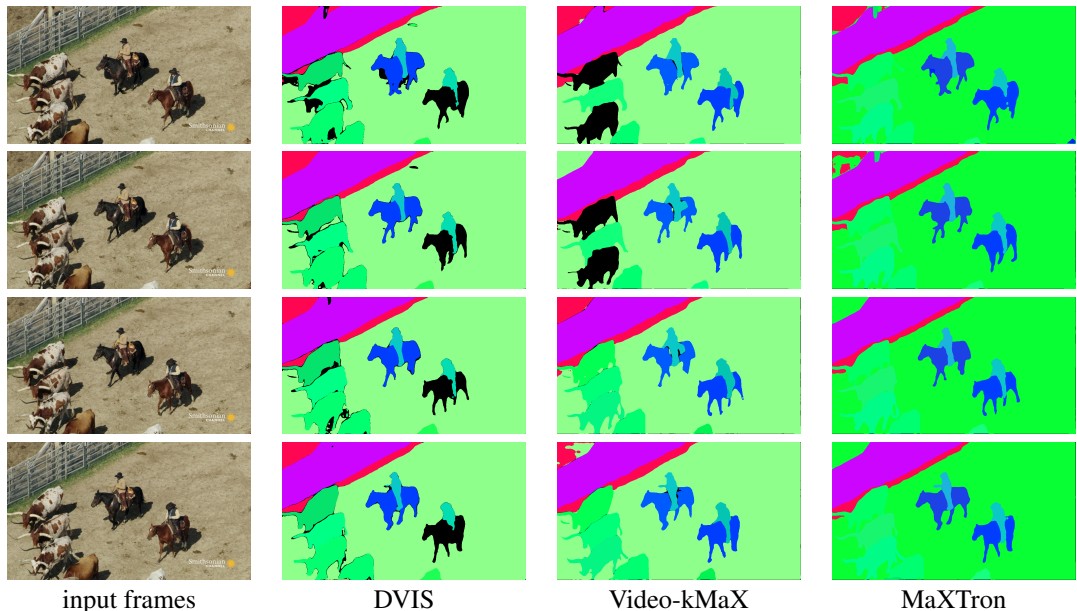

Figure 5: **Qualitative comparisons on videos with unusual viewpoints in VIPSeg.** MaXTron exhibits consistency in prediction even with an unusual view while DVIS and Video-kMaX fail to consistently detect all animals over time.

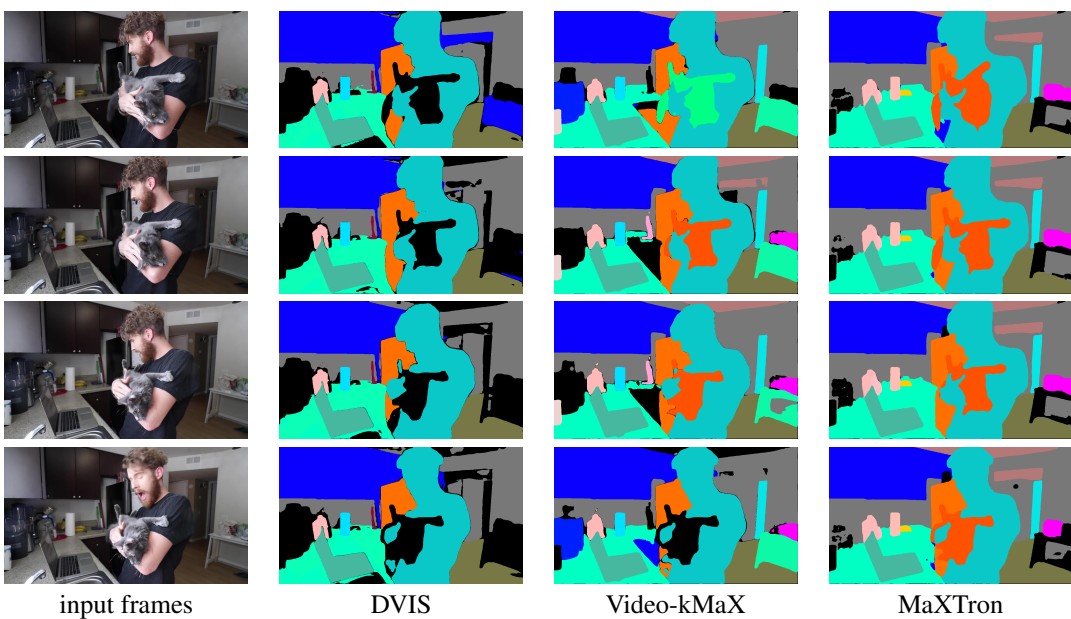

Figure 6: **Qualitative comparisons on videos with complex indoor scenes as background in VIPSeg.** MaXTron accurately segments out the boundary of cat and person with correct classes, while DVIS and Video-kMaX fail to do so.

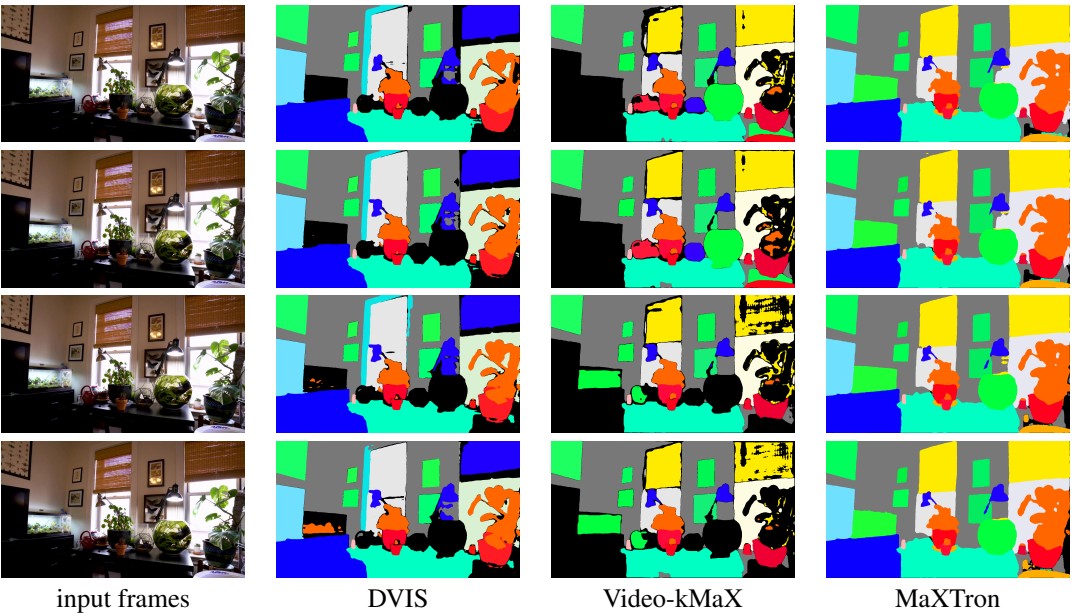

input frames — DVIS — Video-kMaX — MaXTron

Figure 7: **Qualitative comparisons on videos with light and shade in VIPSeg.** MaXTron makes accurate and consistent predictions under different illumination situations. DVIS fails at the junction between light and shade (*e.g.*, the fish tank) while Video-kMaX completely fails at dark places.

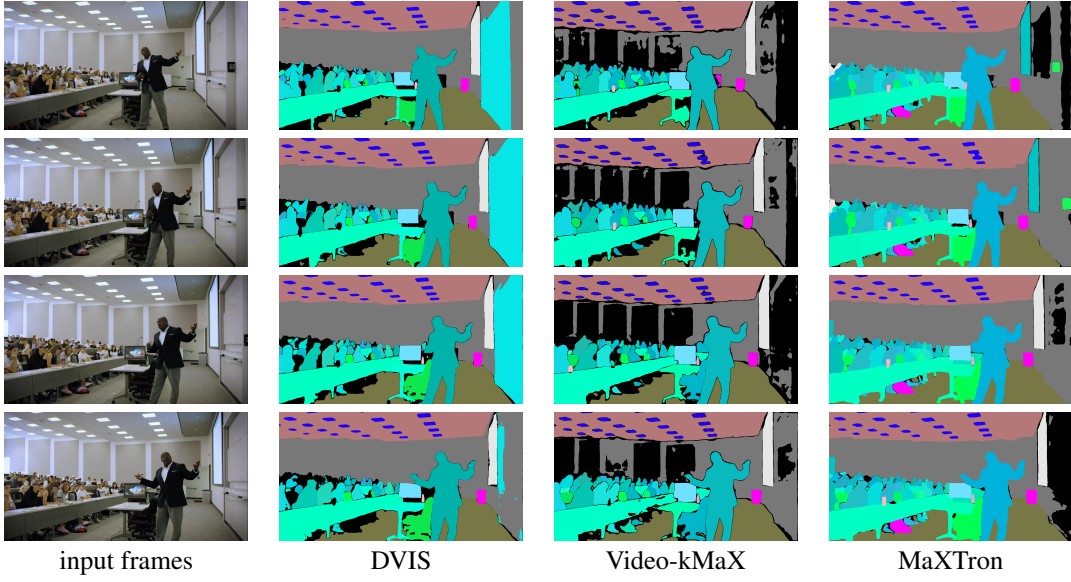

input frames — DVIS — Video-kMaX — MaXTron

Figure 8: **Qualitative comparisons on videos with multiple instances in VIPSeg.** MaXTron detects more instances with accurate boundary. DVIS fails to segment out the crowded humans while Video-kMaX performs badly on the stuff classes.

## F    REBUTTAL

### F.1    MORE AXIAL-TRJECTORY ATTENTION MAP VISUALIZATIONS

We provide more visualizations of the learned axial-trajectory attention maps in Fig. 9, 10, 11 and 12. Concretely, in Fig. 9, we select the basketball in the first frame as the *reference point* and show that our axial-trajectory attention accurately tracks it along the moving trajectory. In Fig. 10, we select the black table in the first frame as the *reference point*. We note that the camera motion is very small in this short clip and the table thus remains static. Our axial-trajectory attention still accurately keeps tracking at the same location as time goes by. In Fig. 11 and 12, we show two failure cases of axial-trajectory attention where the selected *reference point* is not discriminative enough, sometimes yielding inaccurate axial-trajectory. To be specific, in Fig. 11, we select the left light of the subway in the first frame as *reference point*. Though axial-trajectory attention precisely associates its position at the second frame, in the third frame the attention becomes sparse, mostly because that there are many similar 'light' objects in the third frame and the attention dilutes. Similarly, in Fig. 12, we select the head of the human as the *reference point*. Since the human wears a black jacket with a black hat, the selected reference point has similar but ambiguous appearance to the human body, yielding sparse attention activation in the whole human region.

### F.2    FAILURE CASES OF MAXTRON

We provide visualizations of failure cases of MaXTron in Fig. 13 and 14. In general, we observe three common patterns of errors: heavy occlusion, fast moving objects, and extreme illumination.

Specifically, the first challenge is that when there are heavy occlusions caused by multiple close-by instances, MaXTron suffers from ID switching, leading MaXTron to assign inconsistent ID to the same instance. For example, in clip (a) of Fig. 13, the ID of the human in red dress changes between frame 2 and 3, while in clip (b) of Fig. 13 the two humans in the back are recognized as only one human until frame 3 due to the heavy occlusion. The second common error is that in videos containing fast motion, MaXTron suffers from precisely predicting the boundary of the moving object. In clip (c) of Fig. 14, the human's legs are not segmented out in frame 1 and 3. The last common error is that in videos containing extreme or varying illumination, MaXTron might fail to detect the objects thus fails to generate consistent segmentation. In clip (d) of Fig. 14, the objects under the extreme illumination can not be well segmented.

### F.3    DETAILED FIGURES OF TRAJECTORY-ATTENTION AND TEMPORAL-ASPP

We provide figures to illustrate the details of Axial-Trajectory-Attention and Temporal-ASPP in Fig. 15 and 16, respectively. The proposed axial-trajectory attention contains two steps of attention where the first step is to compute the axial-trajectories based on the reference points along $H$-axis or $W$-axis (Eq. 2), and the second step is to conduct temporal attention to aggregate information along the trajectories (Eq. 4). In this way, axial-trajectory attention effectively reasons about the global cross-clip connections. Our Temporal-ASPP module contains three parallel atrous convolutions with different atrous sizes (which are set to (1, 2, 3), respectively) to capture local cross-connections across different time spans.

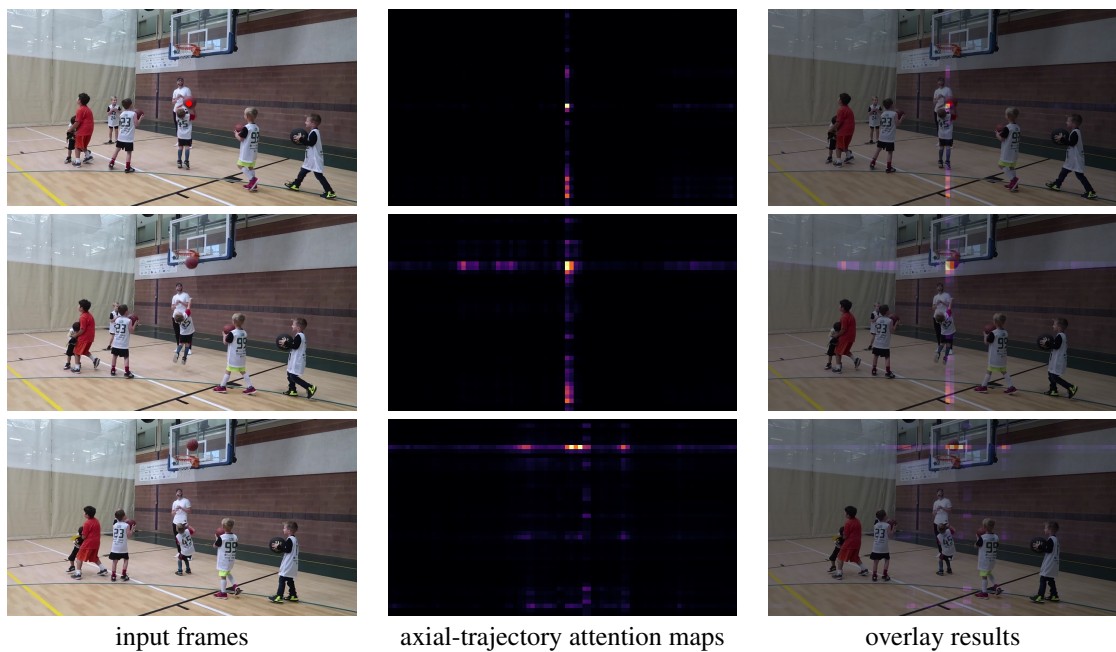

input frames       axial-trajectory attention maps       overlay results

Figure 9: **Visualization of Learned Axial-Trajectory Attention.** In this short clip of three frames depicting the action 'play basketball', the basketball at frame 1 is selected as the *reference point* (mark in red). The axial-trajectory attention is able to accurately track the moving basketball across frames. Best viewed by zooming in.

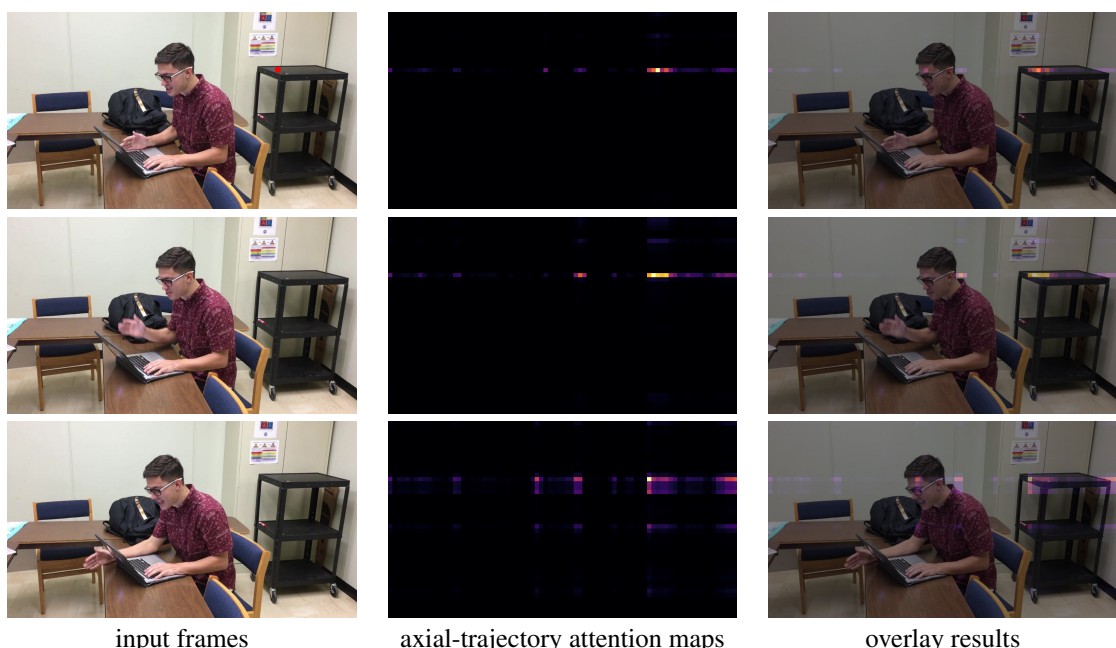

input frames       axial-trajectory attention maps       overlay results

Figure 10: **Visualization of Learned Axial-Trajectory Attention.** In this short clip of three frames depicting a student at class, the right static table at frame 1 is selected as the *reference point* (mark in red). Even though the table remains static across the frames, our axial-trajectory attention is still able to accurately track it. Best viewed by zooming in.

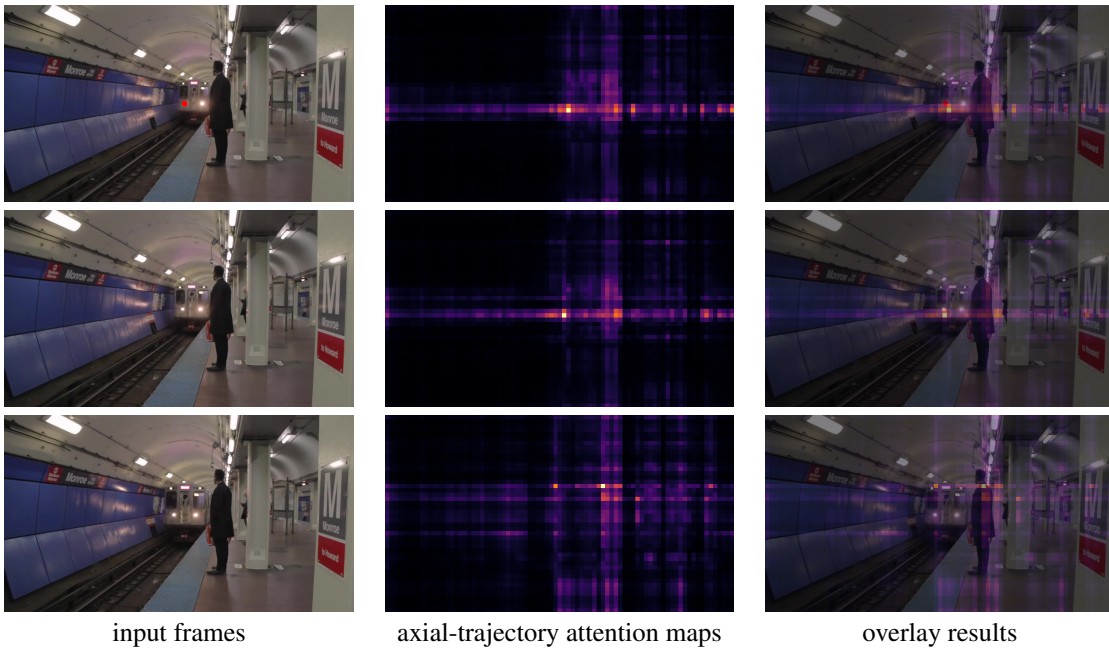

input frames          axial-trajectory attention maps          overlay results

Figure 11: **[Failure mode] Visualization of Learned Axial-Trajectory Attention.** In this short clip of three frames depicting a moving subway, the left front light at frame 1 is selected as the *reference point* (mark in red). While the axial-trajectory attention can still more or less capture the same front light at the frame 2, it gradually loses the focus since there are many similar "light" objects in the clip. Best viewed by zooming in.

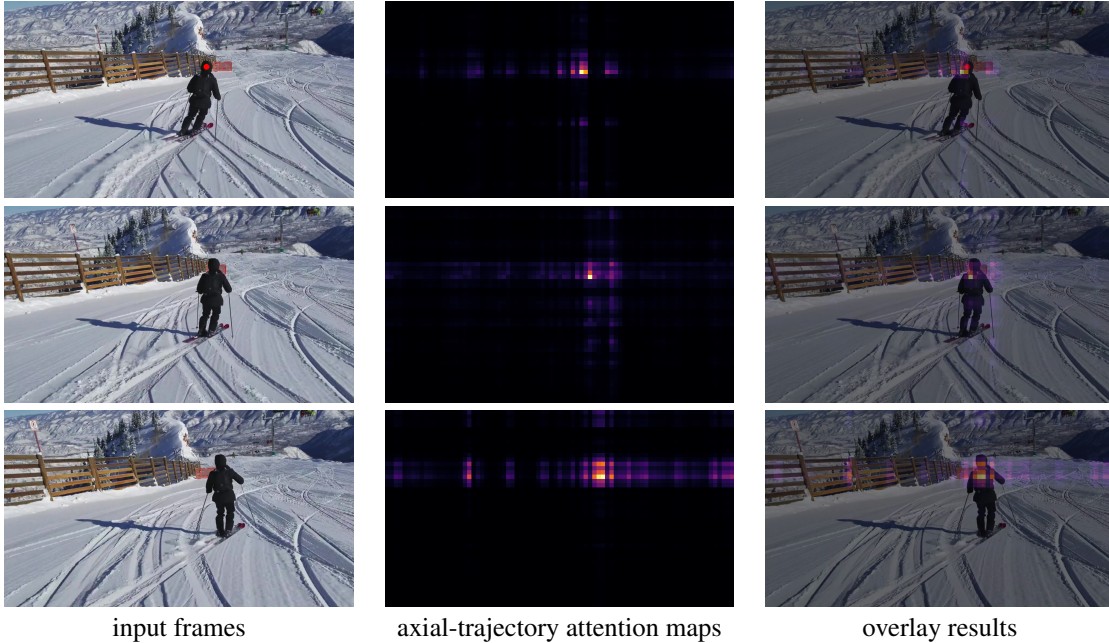

input frames          axial-trajectory attention maps          overlay results

Figure 12: **[Failure mode] Visualization of Learned Axial-Trajectory Attention.** In this short clip of three frames depicting the action 'downhill ski', the head of the human at frame 1 is selected as the *reference point* (mark in red). Since the head and the human body have similar appearance, the axial-trajectory attention becomes diluted among the human body. Best viewed by zooming in.

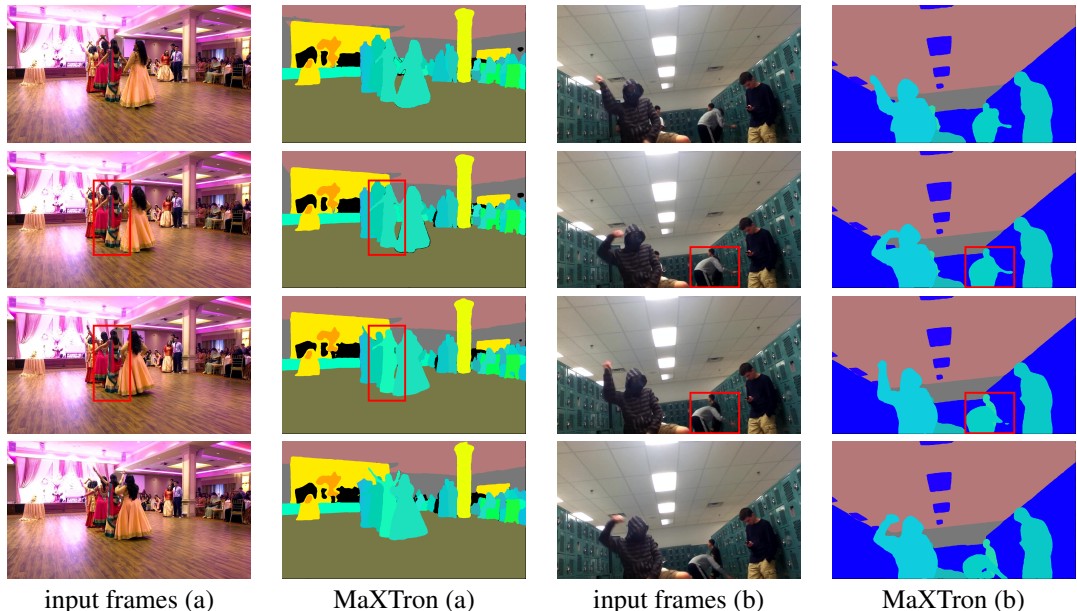

input frames (a)      MaXTron (a)      input frames (b)      MaXTron (b)

Figure 13: **Failure modes caused by heavy occlusion.** MaXTron fails to predict consistent ID for the same instance when there is heavy occlusion. (a) The ID of the human changes between frame 2 and 3 (see red box). (b) The two humans are recognized as only one until frame 3 (see red box). Best viewed by zooming in.

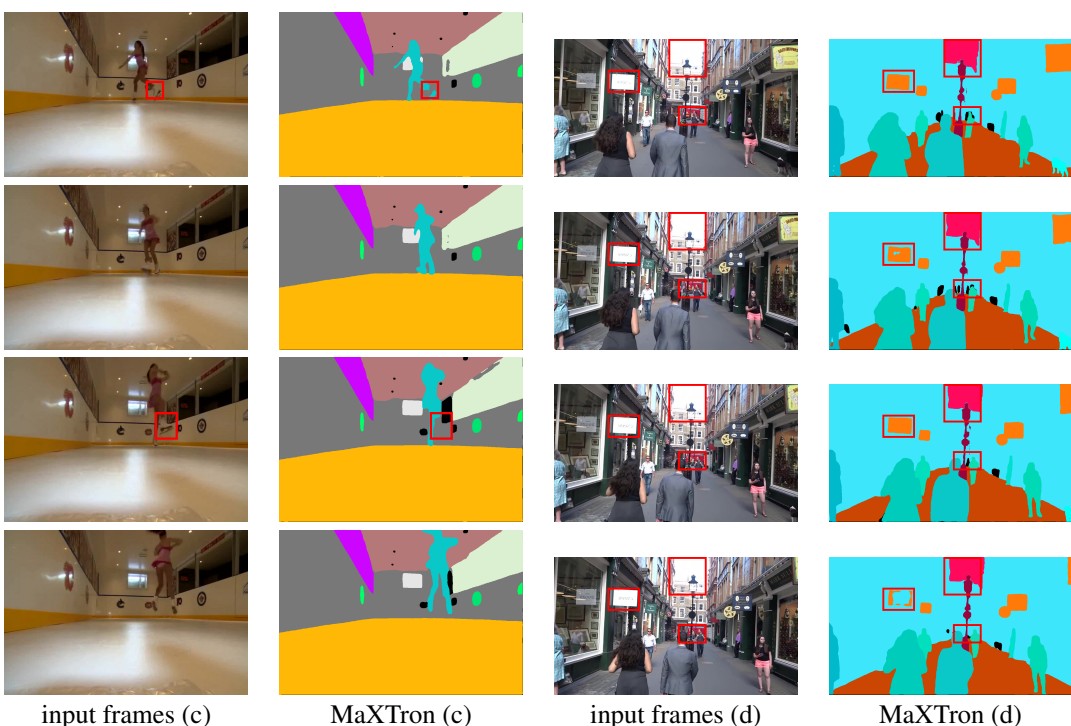

input frames (c)      MaXTron (c)      input frames (d)      MaXTron (d)

Figure 14: **Failure modes caused by fast-moving and extreme illumination scenarios.** MaXTron fails to predict accurate boundary due to the large motion and extreme illumination. (c) The human's leg is not segmented out in frame 1 and 3 (see red box). (d) The objects under extreme illumination can not be well segmented (see red box). Best viewed by zooming in.

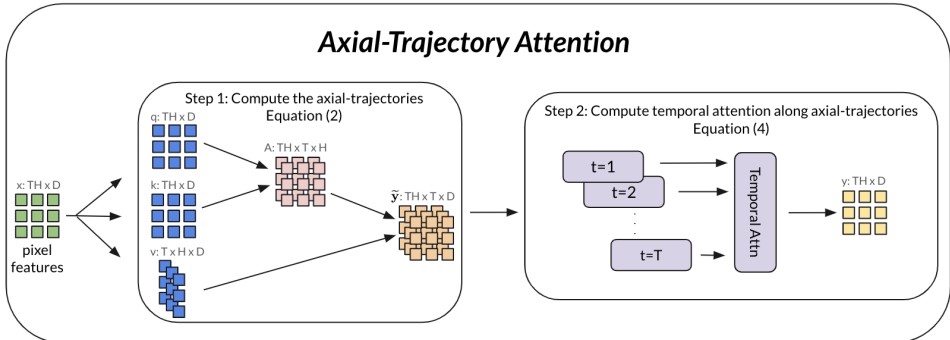

Figure 15: **Illustration of Axial-Trajectory Attention** (only *Height*-axis trajectory attention is shown for simplicity), which includes two steps: computing the trajectories $\widetilde{y}$ along *Height*-axis (Eq. 2) of the dense pixel feature maps $x$ (whose shape is $TH \times D$, where $T$, $H$, and $D$ denote the clip length, input feature height and channels, respectively) and then computing temporal attention along the axial-trajectories (Eq. 4) for effectively capturing the within-clip connections to obtain the updated features $y$.

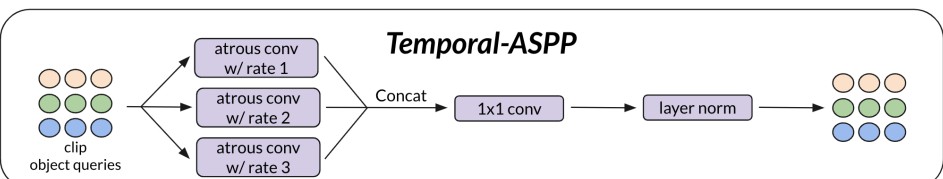

Figure 16: **Illustration of Temporal-ASPP,** which operates on the clip object queries and includes three parallel atrous convolution with different atrous rates to aggregate local temporal cross-clip connections across different time spans followed by 1x1 convolution and layer norm to obtain the final updated clip object queries.

