# OpenReview forum: "MaXTron: Mask Transformer with Trajectory Attention for Video Panoptic Segmentation"
_ICLR.cc/2024/Conference — Submitted to ICLR 2024_

### Official Review · Reviewer_tzju · 2023-10-23

**Soundness:** 2 fair
**Presentation:** 2 fair
**Contribution:** 1 poor
**Rating:** 3
**Confidence:** 5

**Summary:**

This paper tackles the well-known video task, video panoptic segmentation, and presents MaXTron. From the inherent property of the VPS task that comprises other video segmentation tasks such as VSS and VIS, MaXTron can be considered as a general video segmentation framework.
The authors points out a couple of challenges of the video segmentation tasks, and target to alleviate such challenges. Specifically, per-clip segmentation methods (which MaXTron also belongs to) have put efforts in improving inter-clip and intra-clip predictions. In order to improve inter-clip predictions, the authors suggest the Within-Clip Tracking Module, which consists of a stack of multi-scale deformable attention followed by Axial-Trajectory Attentions. For intra-clip association, MaXTron fully leverages the object queries that possess object-level information, and insert into the Cross-Clip Tracking Module that has Trajectory Attention and Temporal ASPP.
Finally, utilizing the presented modules, MaXTron achieves compelling results, demonstrating state-of-the-art accuracy.

**Strengths:**

This paper has a clear structure that ease the readers to follow and understand which components are being used.
The authors points out important problems of the video segmentation tasks and each module is designed with a specific goal.
Combining them all, MaXTron achieves improvements in the accuracy on multiple benchmarks, highlighting the effectiveness of each module.

**Weaknesses:**

The major weakness of this paper is lack of novelty. Each component used in the design of MaXTron is mostly brought from existing literatures or with a subtle modification.
For instance, Multi-level Deformable Attention, Transformer Decoder, and ASPP are brought from previous works.

The used modules with slight changes are 1) Axial-Trajectory Attention and 2) Cross Clip Tracking Module.
The Axial-Trajectory Attention manipulates the set of tokens fed into transformer attentions, which has been widely used in lots of different areas.
Additionally, Cross Clip Tracking Module is a simple modification to VITA, using clip-level outputs instead of frame-level outputs.

As this paper used a number of components that are already proven effective, it is rather expected to see the gain in the accuracy.

**Questions:**

How much are the FLOPs and FPS of MaXTron compared to other methods?
What's the statistical significance, i.e. how many runs were executed for reporting the numbers? Are the numbers mean/median of multiple trials?

---

> ### Author Response · Authors · 2023-11-15
> **To Reviewer tzju (1/2)**
>
> We thank Reviewer tzju for the review, and we address the concerns below.
>
> > W1: Some components are borrowed from literature.
>
> We thank the reviewer for the question. We respectfully disagree with the reviewer and clarify the concerns of each component as below:
>
> >> W1.1: Multi-scale Deformable Attention
>
> Multi-scale Deformable Attention (MSDeformAttn) have been introduced in [1], and used in most recent works. We introduce MSDeformAttn in our within-clip tracking module for a fair comparison to other state-of-the-art methods. Most importantly, we show below that though MSDeformAttn brings improvement to performance, the most significant improvement is still brought by our own proposed axial-trajectory attention.
>
> Firstly, we provide additional results on VIPSeg with ResNet-50 as backbone and ablate on the effectiveness of multi-scale deformable attention and the proposed axial-trajectory attention in terms of VPQ:
>
> MSDeformAttn | axial-trajectory attention | VPQ |
> |:---------:|:----------:|:--------------:|
> X     |    X    |      42.7      |
> $\checkmark$     |    X    |      44.5      |
> X     |    $\checkmark$    |      44.9      |
> $\checkmark$     |    $\checkmark$    |      46.1      |
>
> As can be seen, when compared with the baseline Video-kMaX, adding MSDeformAttn only brings 1.8 VPQ improvement, adding axial-trajectory attention brings 2.2 VPQ improvement. When combining together, we get the final 3.4 VPQ improvement. We also emphasize that Video-kMaX with axial-trajectory attention alone already achieves significant improvement compared to other state-of-the-art methods (Video-kMaX with axial-trajectory attention alone 44.9 vs. current state-of-the-art DVIS 39.2). Please refer to Tab. 5 for more comparisons with different within-clip tracking module designs.
>
> Secondly, for video instance segmentation, please note that Tube-Link already contains MSDeformAttn in their own clip-level segmenter design and we simply add the proposed axial-trajectory attention after their MSDeformAttn. As a result, all performance gains on the near-online setting are brought by the proposed axial-trajectory attention. To be concrete, it brings 0.6 AP, 4.4 $AP^\text{long}$, 5.8 AP improvement when evaluating on Youtube-VIS-21, Youtube-VIS-22 and OVIS dataset with ResNet-50, ResNet-50, and Swin-L as backbone, respectively. Please refer to Tab. 2, 8, 9 and 10 for more comparisons.
>
> In summary, MSDeformAttn does help with the within-clip tracking module, but it is not the core component in it and we do not claim it as our novelty as well. Most importantly, even without MSDeformAttn, our model still brings non-trivial improvement.
>
> >> W1.2: Transformer Decoder
>
> We kindly request the reviewer for clarification on the point that *'Transformer Decoder is borrowed from the literature'.* As far as we are concerned, the transformer decoder is an essential and non-removable component for modern video segmentation models and all other works use it. If the reviewer insists on this point, we kindly request the reviewer for more clarifications, e.g., why the follow-up works should not (and never) use the Transformer Decoder.
>
> >> W1.3: ASPP
>
> ASPP is introduced by [2], and originally operates on the dense pixels in the ***spatial*** domain. As far as we are concerned, we are the first to extend it to the ***temporal*** domain. Motivated by its property in capturing local information, we extend it to operate on the clip object query to reason about the temporal information. This is a non-trivial adaptation of ASPP.
>
> Besides, Tab. 6 (6) in the paper shows that even without ASPP, our usage of trajectory attention on clip object query still brings performance improvement due to the capture of global cross-clip interactions.
>
> [1] Zhu, Xizhou, et al. "Deformable DETR: Deformable Transformers for End-to-End Object Detection". ICLR 2021 (Oral)
>
> [2] Chen, Liang-Chieh, et al. "DeepLab: Semantic Image Segmentation with Deep Convolutional Nets, Atrous Convolution, and Fully Connected CRFs". TPAMI 2017

---

> ### Author Response · Authors · 2023-11-15
> **To Reviewer tzju (2/2)**
>
> > W2: Some components are modified with slight changes.
>
> We thank the reviewer for the question. We respectfully disagree with the reviewer and clarify the concerns of each component as below:
>
> >> W2.1: Axial-Trajectory Attention
>
> We respectfully disagree with the reviewer with the point that *'the Axial-Trajectory Attention manipulates the set of tokens fed into transformer attentions, which has been widely used in lots of different areas'.* As far as we are concerned, ***all*** attention mechanisms manipulate the set of tokens thus it is unfair to blame the design on this. Most importantly, it is non-trivial to adapt trajectory attention to video segmentation task. Though different self-attention mechanisms have been explored a lot in **video classification**, few works try to explore it in **video segmentation** due to the intolerant complexity. Our novel design effectively addresses this key challenge and improves within-clip consistency. Please refer to C1 in common concerns for more discussion on this.
> Finally, we kindly request the reviewer to re-evaluate our contributions. If the reviewer insists on the point that it is trivial to design ***axial-trajectory attention***, we kindly request the reviewer for any reference papers that have ever successfully done so in literature.
>
> >> W2.2: Cross-clip Tracking Module (Comparison with VITA)
>
> Our cross-clip tracking module is not a simple modification from VITA. We clarify the detailed differences compared to VITA below:
>
> 1. VITA introduces an **additional** set of video object queries on top of the existing frame object queries, while MaXTron instead directly manipulates the clip object queries.
>
> 2. VITA contains both an encoder and a decoder to encode the information from frame object queries and decode the information from them via video object queries. As a comparison, MaXTron only employs a simple encoder to aggregate information from clip object queries.
>
> Concretely, the operation flow of VITA is:
> frame query self-attention $\rightarrow$ frame query feed-forward network $\rightarrow$ video query cross-attention to frame query $\rightarrow$ video query self-attention $\rightarrow$ video query feed-forward network
>
> The operation flow of cross-clip tracking module in MaXTron is:
> clip query trajectory-attention $\rightarrow$ clip query Temporal-ASPP
>
> As can be seen, the proposed cross-clip tracking module enjoys a ***much simpler*** pipeline compared to VITA.
>
> 3. VITA uses their proposed video object queries to predict both the whole video mask and class directly, while MaXTron exploits clip object queries to predict clip mask ***separately*** (the final whole video mask is obtained by concatenating the clip masks) and predicts the class by ***weighted mean*** of aligned clip queries, motivated by the fact that the aligned clip queries should have consistent class prediction.
> We argue that the proposed per-clip mask prediction scheme leads to better video mask prediction, since it is challenging to use only one set of video queries to directly predict the mask for the whole video. We provide additional ablation study on this to support our claim by experimenting with MaXTron w/ Video-kMaX with ResNet-50 as backbone on VIPSeg:
>
> mask prediction scheme | VPQ |
> |:---------:|:----------:|
> per-clip   (adopted by us)  |    46.7    |
> per-video (adopted by VITA)    |    46.4    |
>
> 4. Finally, VITA introduces additional similarity loss after introducing their video object queries to better supervise the training, while MaXTron does not need that. Besides, VITA takes frame object queries from the last three layers of the transformer decoder as input, while MaXTron only needs the clip object queries from the last layer.
>
> In the end, we provide experimental results when replacing our cross-clip tracking modules with VITA below (all based on the same Video-kMaX baseline to ensure a fair comparison):
>
> cross-clip tracking design | backbone | input size | GPU days (A100) | params | GFlops (cross-clip tracking module only) | VPQ |
> |:---------:|:----------:|:----------:|:----------:|:----------:|:----------:|:----------:|
> MaXTron     |    ResNet-50    |    $24 \times 769 \times 1345$    |    3.5    |    7.7M    |    32    |    46.7    |
> VITA     |    ResNet-50    |    $24 \times 769 \times 1345$   |    5.2    |    12.9M    |    47    |    46.3   |
>
> As shown in the table, our cross-clip tracking module is ***faster*** in training, ***more lightweight*** in params and FLOPs and achieves ***better*** performance compared to VITA.
>
> > Q1: FLOPs and FPS of MaXTron.
>
> We thank the reviewer for the question. Please refer to common concerns C3 for the FLOPs and FPS comparison.
>
> > Q2: Statistical significance.
>
> We thank the reviewer for the question. Yes, all experiments are conducted 3 times and the reported number is the mean of the results. Please note that we do not report this statistical significance as most works in this literature do not report this in their papers.

---

> ### Comment · Reviewer_tzju · 2023-11-18
> **To authors**
>
> I have read the authors’ response and the reviews from fellow reviewers.
>
> ----
>
> - The main concern that has not fully been addressed is the significance of the claimed contributions. For the past few years, numerous works have been published to address the quadratic computation issue of self-attention. There are many approaches that can handle the CUDA OOM issue such as reducing the scope of attention, decomposing the attention, reducing the number of tokens by taking hierarchical approach, and so on. A lot of those works have already presented such approaches, and they also provide customized CUDA codes that actually makes the model feasible. Compared to those works, I believe the axial-trajectory attention of this paper is much of a naive extension of self-attention: limiting the number of visiting tokens. In order to prove its effectiveness, the authors should have provided thorough analysis of the module and a comparison between temporal extension of existing transformer variants, not only the MSDeformAttn which does only 2D spatial encoding. To list a few, here are some of the references that I believe that could have been simply extended to a spatio-temporal version, and be applied to the VPS task.
>     - Bertasius et al. Is Space-Time Attention All You Need for Video Understanding
>     - Ramachandran et al. Stand-Alone Self-Attention in Vision Models
>     - Hassani et al. Neighborhood Attention Transformer
>     - Beltagy et al. Longformer: The Long-Document Transformer
>     - Xie et al. SegFormer: Simple and Efficient Design for Semantic Segmentation with Transformers
>     - Pan et al. Slide-Transformer: Hierarchical Vision Transformer with Local Self-Attention
>
> Especially, since the first reference can be directly applied, what is the benefit (accuracy & efficiency?) of the proposed module over Bertasius et al?
>
> ----
>
> - Thank you for pointing out that the authors do not claim ASPP as their novelty. However, I do not view the application of ASPP as an enough contribution. It is true that there are not many applications to the dense video pixel-level prediction tasks. However, given that the VPS task is not a significantly popular task, and ASPP is an extremely well-known module in the vision domain, I cannot agree with the authors that it can be considered as a major contribution. Indeed, it is very obvious that ASPP can be seamlessly integrated into any module. I do not mean that this work should not have used the transformer decoder. To clarify, I believe the contributions for the overall architectural design is limited, e.g., using the transformer decoder is to be expected.
>
> -----
>
> - Thanks for providing the comparison to VITA. Despite I understand that the time can be limited for experimenting during the rebuttal period, my remaining concern is that it is only experimented on top of Video-kMaX.
>
> ----
>
> - From the 15 GFlops reduction over VITA, the authors mentioned that the presented module is more computationally efficient. Then, as shown in Table C3, it seems like MaXTron is extremely heavier than Video-kMaX (more than 30% increase).
>     - What’s the GFlops and FPS of other state-of-the-art methods such as TarVIS, VITA, DVIS? Referring to the DVIS paper, DVIS has much less params even than Video-kMaX (which is much lighter than MaXTron).
>         - I believe the FPS comparison can be reported if they are experimented on VIS benchmarks.
>         - Since MaXTron also provides accuracies on VIS datasets, it might be easier and more straight-forward to compare on the VIS benchmarks.

---

> ### Author Response · Authors · 2023-11-19
> **Second Rebuttal to Reviewer tzju (1/3)**
>
> We thank Reviewer tzju for the follow-up reviews, and we address the concerns below.
>
> > T1 The main concern that has not fully been addressed is the significance of the claimed contributions.
>
> We thank the reviewer for the question and the provided references. We are happy to cite and add a brief discussion about them in the revised version.
> Before we start, we would like to clarify that the main aim of this work is not just designing an efficient attention mechanism. Our core objective lies in enhancing the ***tracking ability of clip-level segmenters***, encompassing improvements on both within-clip and cross-clip levels. The introduction of axial-trajectory attention is a non-trivial exploration in advancing this specific pursuit.
>
> > > T1.1 Should compare not only the MSDeformAttn which does only 2D spatial encoding
>
> We note that in our previous rebuttal, we offered the detailed comparison to MSDeformAttn mainly to address your previous concern *"MSDeformAttn is borrowed from literature"* and showed that the proposed axial-trajectory attention brings more significant improvement than MSDeformAttn. We are ***not*** merely comparing with MSDeformAttn, but instead we have carefully compared with the mostly related works to support our claim (***enhancing the tracking ability of clip-level segmenters via axial-trajectory attention***) as shown below:
> - Video segmentation:
>   - TarVIS (for within-clip tracking comparison), VITA (for cross-clip tracking comparison)
> - Video classification
>   -  Divided Space-Time Attention [1].
>
> [1] Bertasius et al. Is Space-Time Attention All You Need for Video Understanding

---

> > ### Author Response · Authors · 2023-11-19
> > **Second Rebuttal to Reviewer tzju (2/3)**
> >
> > > > T1.2 Numerous works have been published to address the quadratic computation issue of self-attention.
> >
> > To address the concerns in time, we systematically analyze the related attention designs, categorized into 3 groups:
> >
> > ***Attention for video segmentation***: As far as we've concerned, for the within-clip tracking design, the most related work to this is TarVIS, which introduces a temporal neck by combining MSDeformAttn and ***Temporal Window Attention***. We have already included discussion on it in our related work and we provide the results of incorporating TarVIS design into Video-kMaX below. As can be observed, it performs worse than the proposed within-clip tracking module with roughly the same amount of computational cost.
> >
> > model | VPQ | GFlops |
> > |:---:|:---:|:---:|
> > Video-kMaX + MSDeformAttn | 44.5 | 432 |
> > Video-kMaX + TarVIS (MSDeformAttn + temporal Window self-attn)  | 44.9 | 476 |
> > Video-kMaX + MaXTron within-clip module (MSDeformAttn + axial-trajectory attn) | 46.1 | 481 |
> >
> > ***Attention for video classification***: We agree with the point that there are many explorations in how to design efficient attention mechanisms for video classification, which are also already included in our related work. However, as a dense prediction task, video segmentation innutively favors trajectory attention due to its ability to capture the object trajectories along the time. We provide the results of incorporating the Divided Space-Time Attention [1] into Video-kMaX below. As can be observed, it does not bring much performance improvement and introduces a considerable computational cost. The observed result is due to the difference in characteristics between the input to video classification and video segmentation. In video classification, the input usually includes 8/16 frames, and each frame is encoded by just a few patches produced by the patch embedding layer. On the other hand, in video segmentation, the input clip contains a very small number of frames (e.g., 2 or 3) while maintaining a large dense pixel feature maps at spatial level. As a result, dividing attention into space and time and conducting them separately still brings a considerable computational cost.
> >
> > model | VPQ | GFlops |
> > |:---:|:---:|:---:|
> > Video-kMaX | 42.7 | 354 |
> > Video-kMaX + divided space-time attn | 43.6 | 430 |
> > Video-kMaX + axial-trajectory attn | 44.9 | 443 |
> >
> > ***Attention for image recognition***: We thank the reviewer for offering the references [2, 3, 5, 6]. We will cite and include a brief discussion on them in the revised version. We agree that it may be promising to extend them for video classification and even video segmentation tasks. However, it is beyond our scope to explore that in this paper.
> >
> > In summary, at within-clip level, it is non-trivial to directly adopt any attention mechanisms designed for video classification or image recognition in order to improve the within-clip tracking ability of the clip-level segmenter. We carefully design our axial-trajectory and show the comparison below. As can be seen, our design achieves similar performance compared to original trajectory attention while making it more efficient (allowing us to use more layers of it) and plausible in video segmentation. More details can be found in Tab. 5 of appendix, where we carefully ablated the within-clip tracking module.
> >
> > model | VPQ | GFlops |
> > |:---:|:---:|:---:|
> > Video-kMaX + MSDeformAttn | 44.5 | 432 |
> > Video-kMaX + MSDeformAttn + trajectory attn * 2 | 45.3 | 494 |
> > Video-kMaX + MSDeformAttn + axial-trajectory attn * 2 | 45.4 | 458 |
> > Video-kMaX + MSDeformAttn + axial-trajectory attn * 4 | 46.1 | 481 |
> >
> > [1] Bertasius et al. Is Space-Time Attention All You Need for Video Understanding
> >
> > [2] Ramachandran et al. Stand-Alone Self-Attention in Vision Models
> >
> > [3] Hassani et al. Neighborhood Attention Transformer
> >
> > [4] Beltagy et al. Longformer: The Long-Document Transformer
> >
> > [5] Xie et al. SegFormer: Simple and Efficient Design for Semantic Segmentation with Transformers
> >
> > [6] Pan et al. Slide-Transformer: Hierarchical Vision Transformer with Local Self-Attention

---

> > > ### Author Response · Authors · 2023-11-19
> > > **Second Rebuttal to Reviewer tzju (3/3)**
> > >
> > > > T2 The contribution for the overall architectural design is limited. VPS task is not a significantly popular task.
> > >
> > > We respectfully disagree with the reviewer and we carefully address the concerns below.
> > > Before we start, we are sorry to hear the subjective comment conerning the VPS task. We argue that VPS (accepted to CVPR 2020 as Oral) is a very new yet important task. Presenting a new challenge to unify video instance and semantic segmentation, it has attracted a rising interest in the community, as evident by two recent competition workshops [1, 2], and extensions to more challening settings [3, 4]. Acknowledging its recent introduction, we contend that the task remains far from being conclusively solved. Hence, we advocate for continued research efforts in this domain.
> > >
> > > [1] Segmenting and Tracking Every Point and Pixel: 6th Workshop on Benchmarking Multi-Target Tracking, ICCV 2021
> > >
> > > [2] The 2nd Pixel-level Video Understanding in the Wild Challenge Workshop, CVPR 2023
> > >
> > > [3] ViP-DeepLab: Learning Visual Perception with Depth-Aware Video Panoptic Segmentation, CVPR 2021
> > >
> > > [4] Waymo Open Dataset: Panoramic Video Panoptic Segmentation, ECCV 2022
> > >
> > > > > T2.1 Temporal-ASPP should not be considered as a major contribution.
> > >
> > > As properly phrased in the submission, we ***never*** claim Temporal-ASPP as our main contributions. Instead, as mentioned in the ***common concerns C1*** and ***T1***, our main contribution lies in successfully extending trajectory attention to the dense video prediction task.
> > > However, the design of Temporal-ASPP based on ASPP is still well-motivated to capture the local cross-clip relations across different time spans. As a result, it contributes to our simple yet effective cross-clip tracking module design as suggested in W2.2.
> > >
> > > > > T2.2 I believe the contributions for the overall architectural design is limited, e.g., using the transformer decoder is to be expected.
> > >
> > > We are again sorry to hear the subjective comment conerning the transformer decoder. We would like to emphasize that the transformer decoder is not our main design. To clarify, we believe every modern video segmentation model has used the transformer decoder, including our baselines (Video-kMaX and Tube-Link). However, as shown in our results, the performance improvements over the baseslines are purely from our designs.
> > >
> > > Specifically, our architectural design, based on an off-the-shelf clip-level segmenter (Video-kMaX or Tube-Link), contains two parts: within-clip tracking module and cross-clip tracking module. This architectural design is quite novel and targets the difficult tracking problem in video segmentation, leading to a significant performance boost over the baseline clip-level segmenter, as shown below.
> > > - 42.7 $\rightarrow$ 46.7 VPQ w/ Video-kMaX R50 on VIPSeg
> > > - 52.7 $\rightarrow$ 57.1 VPQ w/ Video-kMaX ConvNeXt-L on VIPSeg
> > > - 32.1 $\rightarrow$ 37.0 $AP^\text{long}$ w/ Tube-Link R50 on YTVIS-22
> > > - 34.2 $\rightarrow$ 38.9 $AP^\text{long}$ w/ Tube-Link Swin-L on YTVIS-22
> > >
> > > > T3 VITA is only experimented on top of Video-kMaX.
> > >
> > > Regarding the request for additional results, given the remaining limited time, we will try to provide experimental results with VITA on top of Tube-Link on Youtube-VIS in the next coming post.
> > >
> > > > T4 Computational cost and efficiency of MaXTron.
> > >
> > > > > T4.1 MaXTron is extremely heavier than Video-kMaX
> > >
> > > With more than 30% increase of GFlops, we improve the performance by a remarkable margin from 42.7 to 46.1 (using only within-clip tracking module). We do not focus on efficient attention or real-time attention, and most importantly, we ***never*** claim that our work is the most efficient method in GFlops. Other existing works also introduce additional computations as well.
> > >
> > > > > T4.2 GFlops and FPS of other state-of-the-art methods such as TarVIS, VITA, DVIS
> > >
> > > Regarding the request for additional results, we will try to provide GFlops and FPS comparison to TarVIS, VITA and DVIS in VIS in the next coming post.
> > >
> > > But, we would like to emphasize that we ***never*** claim our method to be the more efficient in GFlops and FPS than those methods. We are curious to hear from the reviewer regarding the reason to request more results in the very late stage of the review process. Particularly, what will those results support any of the contribution claims that were made in our submission?
> > >
> > > > > T4.3 MaXTron is not parameter-efficient
> > >
> > > We would like to emphasize again that we ***never*** claim our method to be the most parameter-efficient than any other method. Given the "plug-in" nature of our method, we focus on performance improvement in the work.

---

> > > > ### Author Response · Authors · 2023-11-20
> > > > **Second Rebuttal to Reviewer tzju (4/3)**
> > > >
> > > > We provide the requested additional experimental results below.
> > > >
> > > > > T3 VITA is only experimented on top of Video-kMaX.
> > > >
> > > > As suggested, we extend VITA on top of Tube-Link for video instance segmentation. All experiments are based on the same (Tube-Link + axial-traject attn) checkpoint to ensure a fair comparison. Similarly, we observe that the proposed MaXTron cross-clip tracking design performs better than VITA on YTVIS-22.
> > > >
> > > > model | $AP^\text{long}$ on YTVIS-22 | GFlops | FPS |
> > > > |:---:|:---:|:---:|:---:|
> > > > Tube-Link + axial-traject attn + VITA cross-clip | 36.7 | 2634 | 11.3 |
> > > > Tube-Link + axial-traject attn + MaXTron cross-clip | 37.0 | 2594 | 12.7 |
> > > >
> > > > > > T4.2 GFlops and FPS of other state-of-the-art methods such as TarVIS, VITA, DVIS
> > > >
> > > > The table below provides a summary of the $AP^\text{long}$ on YTVIS-22, GFlops, and FPS comparison with other state-of-the-art methods. GFlops is computed given an input video of size $36 \times 360 \times 640$. FPS is reported based on the inference speed on YTVIS with a single TITAN RTX GPU. As shown in the table, MaXTron remarkably enhances the performances of Tube-Link, while demonstrating comparable GFlops and FPS with other state-of-the-art methods.
> > > >
> > > >
> > > > model | $AP^\text{long}$ on YTVIS-22 | input size | GFlops | FPS |
> > > > |:---:|:---:|:---:|:---:|:---:|
> > > > VITA | 31.9 | $36 \times 360 \times 640$ | 2165 | 24.7 |
> > > > TarVIS | 33.8 | $36 \times 360 \times 640$ | 2781 | 11.8 |
> > > > DVIS Online | 31.2 | $36 \times 360 \times 640$ | 2136 | 23.6 |
> > > > DVIS Offline | 35.9 | $36 \times 360 \times 640$ | 2226 | 17.3 |
> > > > Tube-Link | 32.1 | $36 \times 360 \times 640$ | 2219 | 19.4 |
> > > > Tube-Link + axial-traject attn (MaXTron near-online) | 36.5 | $36 \times 360 \times 640$ | 2530 | 16.8 |
> > > > Tube-Link + axial-traject attn + MaXTron cross-clip (MaXTron offline) | 37.0 | $36 \times 360 \times 640$ | 2594 | 12.7 |

---

### Official Review · Reviewer_LGcv · 2023-10-25

**Soundness:** 3 good
**Presentation:** 2 fair
**Contribution:** 2 fair
**Rating:** 6
**Confidence:** 3

**Summary:**

This work proposes a novel panoptic segmentation method, namely MaXTron, which enhances temporal consistency by the proposed within-clip and cross-clip tracking modules. Axial-trajectory attention is the essential component of the introduced tracking modules, which aims at associating objects meanwhile reducing computational complexity.Experimental results have shown state-of-the-art performance on video segmentation benchmarks.

**Strengths:**

It sounds interesting to conduct in-clip tracking and cross-clip tracking via axial-trajectory attention.

Association with non-overlapping clips is more efficient than previous overlapping-based methods.

The results are promising.

**Weaknesses:**

The writing of sec.3 (method) should be improved. It‘s a bit confusing about the implementation details.

Besides the performance, it is suggested to provide the cost, e.g. training cost and inference speed, of integrating the proposed model into existing methods.

Besides the overall performance, a deeper analysis is expected. For instance, how does the association capability improve after integrating the proposed modules into an off-the-shelf method?

**Questions:**

see weakness

---

> ### Author Response · Authors · 2023-11-15
> **To Reviewer LGcv**
>
> We thank Reviewer LGcv for the review, and we address the concerns below.
>
> > W1: Writing should be improved. confusing about the implementation details.
>
> We thank the reviewer for pointing this out and for the suggestions. Please refer to Q3, W1 and W2 in response to Reviewer TBSQ in case there are any confusions regarding axial-trajectory attention, trajectory attention and Temporal-ASPP, respectively.
>
> Additionally, as promised in our abstract, we will open-source all our implementations and checkpoints, allowing the community to scrutinize the details (for both training and inference).
>
> > W2: Provide the cost, e.g. training cost and inference speed
>
> We thank the reviewer for pointing this out. Please refer to common concerns C3 for detailed training cost and inference speed.
>
> > W3: Deeper analysis of how the association capability is improved by the proposed modules.
>
> We thank the reviewer for pointing this out. The association capability is improved with two modules: within-clip tracking and cross-clip tracking.
>
> In the within-clip tracking module, MaXTron effectively exploits axial-trajectory attention to reason about the pixel-level temporal association. Concretely, axial-trajectory attention finds the probabilistic path of each pixel along time as shown in Fig. 1. It then aggregates information along this probabilistic path. Intuitively, this helps to capture pixel correspondence and avoids inconsistent prediction within a clip. As a result, the within-clip tracking module brings **3.4** and **3.5** VPQ improvement on VIPSeg, with ResNet50 and ConvNeXt-L, respectively (see Tab. 1 in the paper).
>
> In the cross-clip tracking module, MaXTron exploits trajectory attention along with Temporal-ASPP to capture both the global and local cross-clip connections. Intuitively, the per-clip prediction is not perfect and simply associating them with Hungarian Matching as done in most existing online/near-online methods might lead to incorrect matching. This might further lead to problems such as ID switching or miss detection. By introducing the cross-clip tracking module, we aim at aligning the ***clip object queries*** more precisely by modeling the connections between them. As a result, our offline method with the proposed cross-clip tracking module is able to provide more consistent prediction at video-level and thus brings an additional **0.6** and **0.9** VPQ improvement on VIPSeg, with ResNet50 and ConvNeXt-L, respectively (see Tab. 1 in the paper).

---

> > ### Comment · Area_Chair_F3fT · 2023-11-19
> >
> > Dear Reviewer,
> >
> > The author has provided responses to your questions and concerns. Could you please read their responses and ask any follow-up questions, if any?
> >
> > Thank you!

---

### Official Review · Reviewer_tzTu · 2023-10-25

**Soundness:** 4 excellent
**Presentation:** 3 good
**Contribution:** 3 good
**Rating:** 6
**Confidence:** 4

**Summary:**

In this paper, the authors proposed trajectory attention based mask transformer for video panoptic segmentation. Specifically, two types of tracking modules (within-clip and cross-clip tracking) are proposed to improve the temporal consistency by leveraging trajectory attention. The within-clip tracking module, an axial-trajectory attention is proposed for effectively computing the trajectory attention for tracking dense pixels sequentially along the height- and width-axes, while the cross-clip tracking module is used to capture the long-term temporal connections by applying trajectory attention to object queries. The experimental shows that the proposed solution is able to help boost the performance of existing solutions (e.g., Video-kMax and Tube-Link) on multiple datasets.

**Strengths:**

1. The proposed solution sounds solid. (1) Using trajectory attention to force the model pay more attention spatially and temporally on trajectories (maybe simply on pixel trajectories) while doing video segmentation sound solid in theory. The attention should be able to provide extra useful information to the model. (2) Splitting the trajectory attention along different axes (horizontal and vertical) indeed helps reduce the complexity while calculating attention.
2. The experimental results on multiple datasets and models prove that the proposed solution works in varying application scenarios.
3. This paper is well-organized, which help readers easy to read and understand. Expecially, there are more implementation details and results reported in the appendix, which helps readers better understand their work and the performance.

**Weaknesses:**

1. It will be better to report some failure cases. It will be helpful if the authors could report some failure cased that caused by applying the proposed MaxTron. In this case, readers will better understand their work and the performance, which may inspire more ideas along this direction.
2. The proposed solution sounds like an add-on to the existing solutions, which was inspired by other works (e.g., Patrick et al. 2021). The novelty may be incremental.

**Questions:**

1. What if we change the number of frames within one video clip? Is there any positive / negative impact on the model performance? Is there any guidances (or suggestion) of how many frames should be selected while spliting the video?
2. Does the proposed solution perform differently if we (1) process all continuous frames or (2) only process key frames (with some down-sampling temporally)? The later operation will speed up motions in videos.

---

> ### Author Response · Authors · 2023-11-15
> **To Reviewer tzTu**
>
> We thank Reviewer tzTu for the review, and we address the concerns below.
>
> > Q1: What if we change the number of frames within one video clip?
>
> We thank the reviewer for bringing up this question. We observe performance change when varying the number of frames within one video clip. Most likely, it is due to the fact that the off-the-shelf clip-level segmenter itself performs differently with different clip sizes. Concretely, for Video-kMaX and MaXTron w/ Video-kMaX on VIPSeg with ResNet50 as backbone in terms of VPQ, we have:
>
> clip size | Video-kMaX | MaXTron online | MaXTron offline |
> |:---------:|:----------:|:--------------:|:---------------:|
> 2     |    42.7    |      46.1      |       46.7      |
> 3     |    42.1    |      45.1      |       45.5      |
> 4     |    41.4    |      44.2      |       44.7      |
>
> As we can see, with the increase of clip size, Video-kMaX performance gradually decreases. However, both our MaXTron online (w/ within-clip tracking module) and MaXTron offline (w/ within-clip + cross-clip tracking module) models bring steady and consistent improvements. We hypothesize that the performance difference of Video-kMaX is due to the lack of a temporal module in its framework and the temporal connections are forced to be modeled by the transformer decoder only. As a result, with the increase of clip size, the transformer decoder can not handle the features properly, leading to the decrease in performance. Our axial-trajectory attention alleviates this problem and achieves slightly larger improvement when clip size = 2. Besides, our cross-clip tracking module is less sensitive to clip size as it directly learns to model video-level input.
>
> > Q2: Does the proposed solution perform differently if we (1) process all continuous frames or (2) only process key frames (with some down-sampling temporally)?
>
> We thank the reviewer for bringing up this question. We would like to clarify the challenges of dense prediction is to label all pixels in the video, and it is very challenging to only process key frames and propagate the key frame results to other frames during testing.
>
> However, as suggested in C2 in common concerns where we discuss possible failure cases, one main challenge that still remains for MaXTron is to model fast-moving objects. As a result, we hypothesize that MaXTron will perform worse if the video itself contains fewer frames, which might be caused by down-sampling with a large factor in the temporal domain. This is partially supported by our experiments where we train the model with discontinuous sampled frames (specified by the frame sampling range below):
>
> frame sampling range | VPQ |
> |:---------:|:----------:|
> $\pm1$     |    46.1    |
> $\pm2$     |    45.9    |
> $\pm3$     |    45.8    |
> $\pm5$     |    45.3    |
> $\pm10$     |    43.9    |
>
> Above experiments are done with MaXTron w/ Video-kMaX on VIPSeg with ResNet-50 as backbone, clip size = 2 and are evaluated in terms of VPQ. As shown in the table, training with continuous frames brings the best performance. While slightly increasing the sampling range does not affect the performance a lot, increasing it to $\pm10$ greatly hampers the learning of the within-clip tracking module.
>
> > W1: Report some failure cases.
>
> We thank the reviewer for pointing this out. Please refer to common concerns C2 for the discussions on common error patterns of MaXTron.

---

> > ### Comment · Area_Chair_F3fT · 2023-11-19
> >
> > Dear Reviewer,
> >
> > The author has provided responses to your questions and concerns. Could you please read their responses and ask any follow-up questions, if any?
> >
> > Thank you!

---

> > ### Comment · Reviewer_tzTu · 2023-11-20
> >
> > Thank you so much for the authors' responses. The extra results are helpful, and the analysis is sound reasoning. I do not have extra questions. Thanks.

---

> > > ### Author Response · Authors · 2023-11-20
> > > **Thanks to Reviewer tzTu**
> > >
> > > We sincerely thank the reviewer for the valuable review and feedback.

---

### Official Review · Reviewer_TBSQ · 2023-10-28

**Soundness:** 3 good
**Presentation:** 3 good
**Contribution:** 2 fair
**Rating:** 5
**Confidence:** 4

**Summary:**

In this paper, the authors mainly study the clip-level video panoptic segmentation. They propose a new framework using Mask Xformer with trajectory attention, named MaXTron. It includes within-clip and cross-clip tracking modules, to use trajectory attention. The experimental results show the effectiveness of their proposed model.

**Strengths:**

This paper is well-written and easy to follow. The idea of using trajectory information to help the segmentation and decompose the attention into height and width in two directions, greatly reducing the computational complexity. They have done comparison and ablation studies to validate their proposed components.

**Weaknesses:**

The figures might not be easy to follow. For example, in Fig. 3, they show H and W-axis attention maps of one point. It would be much better, if they also show how to get the probabilistic path of a point between frames and what the whole attention maps for static and dynamic points. Besides, the authors should show the details in trajectory attention module and temporal ASPP in Fig. 4 and it can help readers to understand.

The main contribution of this work is the trajectory based within-clip and cross-clip module, which might be limited and insufficient for this conference, even if the authors could clearly introduce their modules using Fig. 3 and 4 after revision.

In the experiment, the authors are suggested to add some examples to show the attention maps and how to get the trajectories or the trajectories might not be perfect.

**Questions:**

The main contribution of this work is the trajectory-based within-clip and cross-clip modules, which might be limited and insufficient for this conference, even if the authors could clearly introduce their modules using Fig. 3 and 4 after revision.

In the experiment, the authors are suggested to add some examples to show the attention maps and how to get the trajectories or the trajectories might not be perfect.

It would be much better if they also showed how to get the probabilistic path of a point between frames and what the whole attention maps for static and dynamic points.

---

> ### Author Response · Authors · 2023-11-15
> **To Reviewer TBSQ (1/2)**
>
> We thank Reviewer TBSQ for the review, and we address the concerns below.
>
> > Q1: Main contribution?
>
> As properly phrased in the submission, we ***never*** claim trajectory-attention as our main contributions. Instead, as mentioned in the ***common concerns C1***, our contribution lies in the fact that we are the first to successfully adapt trajectory-attention to the dense pixel prediction tasks, by designing ***axial-trajectory-attention*** for tracking pixel movement in *H* -axis and *W*-axis separately to model the within-clip temporal cues as well as exploiting trajectory-attention on ***clip object queries*** to model the cross-clip temporal connections.
>
> To the best of our knowledge, none of the existing works have successfully explored trajectory-attention for dense prediction tasks and very few works ever explored temporal attention for dense video prediction. As shown in the experimental results, our attempt has successfully led to a significant boost. We kindly request the reviewer to re-evaluate our contributions. If the reviewer insists on the point that it is ***limited*** and ***insufficient*** to design trajectory-based within-clip and cross-clip modules for dense prediction, we kindly request the reviewer to elaborate more on the reasons.
>
> > Q2: Examples of attention maps and how to get the trajectories or the trajectories might not be perfect.
>
> >> Q2.1 More examples of attention maps
>
> Please see the revised version for more examples of attention maps of selecting either dynamic or static points as reference points. Concretely, please refer to Fig. 9, 10, 11, 12 and Sec. F. 1 in Appendix for the detailed discussion.
>
> >> Q2.2 How to get the trajectories?
>
> Please see Q3 for detailed explanation.
>
> >> Q2.3 Trajectories might not be perfect.
>
> Please see the revised version for failure cases (Fig. 13 and 14). Besides, as suggested by the discussions in C2 in common concerns, the trajectories might also fail when the selected reference points are not discriminative enough.
>
> > Q3: How to get the probabilistic path of a point between frames?
>
> **Overview:** We first provide an overview of how to get the probabilistic path. Conceptually, the affinity logits of pixels across different frames are computed within the process of axial-trajectory attention. As a result, simply exploiting the computed attention map can bring us the visualizations of the probabilistic path. For your convenience, we now explain it in detail below.
>
> We take *H*-axis trajectory attention for example. We first reshape the feature map extracted by the backbone $F \in \mathbb{R}^{\mathit{T} \times \mathit{D} \times \mathit{H} \times \mathit{W}}$ into $F_{h} \in \mathbb{R}^{\mathit{W} \times \mathit{TH} \times \mathit{D}}$ before conducting *H*-axis trajectory attention. In this way, we obtain a sequence of $\mathit{TH}$ pixel features $x_{th} \in \mathbb{R}^{D}$.
>
> Then the *H*-axis trajectory attention can be divided into two steps.
>
> 1. **Trajectory along *H*-axis (Eq. (2) in the paper):** Given a reference point at a specific time-height th position, we compute its possible time-height locations at each frame separately followed by extracting per-frame information (i.e., trajectories) with the computed probability.
>
> 2. **Temporal attention along the trajectory (Eq. (4) in the paper):** Given the trajectories, we aggregate the information along them using 1D time attention to reason about temporal connections.
>
> The construction of the probabilistic path lies in step 1. Concretely, we use linear projection to project all features $x_{th}$ into their corresponding queries, keys and values. Then for a given new time $t'$ (i.e., the frame index), the trajectory tokens are computed by (Eq. (2) in the paper): $\widetilde y_{tt'h} = \sum_{h'}v_{t'h'} \cdot \frac{\exp{\langle q_{th},k_{t'h'} \rangle}}{\sum_{\overline{h}}\exp{\langle q_{th}, k_{t'\overline{h}}\rangle}}$.
>
> Intuitively, the operation in the numerator $\langle q_{th},k_{t'h'} \rangle$ yields an attention map $A \in \mathbb{R}^{\mathit{T} \times \mathit{H}}$, which computes the similarity logits of the reference point th and pixel features at all other possible TH locations $t'h'$.
>
> As a result, to draw the *H*-axis trajectory attention map across frames, we first pick a reference point th (e.g. the basketball in Fig. 1) followed by computing the attention map $A$ as above. We can then set the new time point to $t+1$, $t+2$, ... $t+k$ and retrieve the corresponding position at *H*-axis iteratively. By putting them together, we get the trajectory path along *H*-axis. Similarly, the same operation is conducted for the *W*-axis to obtain the trajectory path along the *W*- axis. As explained in the caption of Fig. 1, for the purpose of visualization, we multiply the *H*-axis and *W*-axis trajectory attentions to visualize the trajectory of the reference point over time (i.e., a bright point corresponds to a high attention value in both the H- and W-axis trajectory attention).

---

> > ### Author Response · Authors · 2023-11-15
> > **To Reviewer TBSQ (2/2)**
> >
> > > W1: Lack of details of the trajectory attention module.
> >
> > We thank the reviewer for pointing this out. We believe that we have provided enough details of the trajectory attention module. However, for your reference, we introduce again the trajectory attention module below. Finally, we sincerely seek more detailed and constructive feedback from the reviewer, particularly, which part confuses you. We also kindly invite the reviewer to read the original published trajectory attention paper at NeurIPS 2021 (Oral) [1].
> >
> > The cross-clip tracking module takes the object queries of all clips as input, which are already associated with target objects (e.g., the target 'person' or 'car') within each clip. Intuitively, the target objects might appear in different clips (e.g., due to occlusion), but the corresponding object queries at different clips should be similar. Therefore, we expect the model to be able to extract this cross-clip information from the object queries. Motivated by this, we apply trajectory attention to the clip object queries. To be specific, for a given object query at clip *k*, we conduct trajectory attention in two steps:
> >
> > 1. **Trajectory Computation (Eq. (5) in the paper):** Search for its possible corresponding query at other clips independently and aggregate this information with the computed probability to form the query trajectories.
> > 2. **Temporal Attention along the Trajectory (Eq. (6) in the paper):** Aggregate information along the query trajectories to reason about global cross-clip interactions.
> >
> > Note that compared to the ***axial-trajectory attention*** in within-clip tracking module which is applied on the dense pixels, here the ***trajectory attention*** aims at extracting the global cross-clip (i.e., all clips) information by tracking the queries instead.
> >
> > > W2: Lack of details of the Temporal-ASPP module.
> >
> > We thank the reviewer for pointing this out. Similarly, we believe that we have provided enough details of the proposed Temporal-ASPP module. However, for your reference, we introduce it again below:
> >
> > Our Temporal-ASPP extends the classical ASPP to the temporal domain. Intuitively, with trajectory attention capturing the global cross-clip interactions, we exploit Temporal-ASPP to capture more local cross-clip interactions (i.e., within the neighboring clips). Concretely, our Temporal-ASPP contains three parallel temporal atrous convolutions with different atrous rates set to (1, 2, 3), as ablated in appendix Tab. 6 (b). The proposed Temporal-ASPP is also operated on clip object queries to capture local cross-clip connections with different time spans.
> >
> > As a result, we iteratively stack trajectory attention and the Temporal-ASPP to construct our cross-clip tracking module to reason for both global and local cross-clip information. They are stacked 6 times based on Video-kMaX for VPS and 4 times based on Tube-Link for VIS.
> >
> > > W3: Fig. 3 and 4 should be improved.
> >
> > We thank the reviewer for the suggestion. Please see the revised version for the additional figures (Fig. 15 and 16) illustrating the details of axial-trajectory attention and Temporal-ASPP.
> > We note that MSDeformAttn is a common but complex operation, which is typically illustrated as an MSDeformAttn module, and it is beyond our scope to well-illustrate it in our figures.
> >
> > [1] Patrick, Mandela, et al. "Keeping Your Eye on the Ball: Trajectory Attention in Video Transformers". NeurIPS 2021 (Oral)

---

> > > ### Comment · Area_Chair_F3fT · 2023-11-19
> > >
> > > Dear Reviewer,
> > >
> > > The author has provided responses to your questions and concerns. Could you please read their responses and ask any follow-up questions, if any?
> > >
> > > Thank you!

---

> > > > ### Comment · Reviewer_TBSQ · 2023-11-20
> > > > **Response to authors.**
> > > >
> > > > I have carefully read other reviews and the authors' feedback. Thanks for the great length and detailed feedback. For questions about the lack of details of the trajectory attention and Temporal-ASPP module, the authors are suggested replacing Fig. 3 and 4 with Fig. 15 and 16, since Fig. 15 and 16 have more details not just a bounding box in Fig. 3 and 4. However, after reading the all responses, I decided to hold the rating mainly on the novelty and contributions. This work has extended or stepped forward on the current method, like trajectory attention for video or ASPP, in terms of reducing computer costs. But as pointed out by reviewer tzju, the authors might consider other quadratic computation issues of self-attention.

---

> > > > > ### Author Response · Authors · 2023-11-20
> > > > > **Second Rebuttal to Reviewer TBSQ**
> > > > >
> > > > > We express our gratitude to Reviewer TBSQ for the follow-up reviews and valuable suggestions. Before delving into the remaining concerns, we wish to kindly remind Reviewer TBSQ that we have made dedicated efforts to address the concerns raised by Reviewer tzju regarding *other quadratic computation issues of self-attention.* Nonetheless, to ensure comprehensive coverage, we carefully reiterated these points below for your convenience. Additionally, we eagerly await feedback from you.
> > > > >
> > > > > > S1 This work has extended or stepped forward on the current method, like trajectory attention for video or ASPP, in terms of reducing computer costs.
> > > > >
> > > > > We have discussed the main target of this work in ***T1*** in the "Second Rebuttal to Reviewer tzju (1/3)". We copy them here for your convenience:
> > > > >
> > > > > We would like to clarify that the main aim of this work is not just designing an efficient attention mechanism. Our core objective lies in enhancing the ***tracking ability of clip-level segmenters***, encompassing improvements on both within-clip and cross-clip levels. The introduction of axial-trajectory attention is a non-trivial exploration in advancing this specific pursuit.
> > > > >
> > > > > > S2 The authors might consider other quadratic computation issues of self-attention.
> > > > >
> > > > > We have discussed this issue in ***T1.2*** in the "Second Rebuttal to Reviewer tzju (2/3)". For your convenience, we briefly summarize them again below:
> > > > >
> > > > > We systematically analyze the related attention designs, categorized into 3 groups:
> > > > >
> > > > > ***Attention for video segmentation***: As far as we've concerned, for the within-clip tracking design, the most related work to this is TarVIS, which introduces a temporal neck by combining MSDeformAttn and ***Temporal Window Attention***. We provide the results of incorporating TarVIS design into Video-kMaX below and show that it performs worse than the proposed within-clip tracking module.
> > > > >
> > > > > model | VPQ | GFlops |
> > > > > |:---:|:---:|:---:|
> > > > > Video-kMaX + MSDeformAttn | 44.5 | 432 |
> > > > > Video-kMaX + TarVIS (MSDeformAttn + temporal Window self-attn)  | 44.9 | 476 |
> > > > > Video-kMaX + MaXTron within-clip module (MSDeformAttn + axial-trajectory attn) | 46.1 | 481 |
> > > > >
> > > > > ***Attention for video classification***: There are many explorations in how to design efficient attention mechanisms for video classification as already included in our related work. However, as a dense prediction task, video segmentation innutively favors trajectory attention due to its ability to capture the object trajectories along the time. We provide the results of incorporating the Divided Space-Time Attention [1] into Video-kMaX below and show that it does not bring much performance improvement and introduces a considerable computational cost, which is caused by the innate difference between the video input to the two tasks. We refer to ***T1.2*** for detailed discussion.
> > > > >
> > > > > model | VPQ | GFlops |
> > > > > |:---:|:---:|:---:|
> > > > > Video-kMaX | 42.7 | 354 |
> > > > > Video-kMaX + divided space-time attn | 43.6 | 430 |
> > > > > Video-kMaX + axial-trajectory attn | 44.9 | 443 |
> > > > >
> > > > > ***Attention for image recognition***: We will cite and include a brief discussion in comparison to attention for image recognition [2, 3, 5, 6].  We agree that it may be promising to extend them for video classification and even video segmentation tasks. However, it is beyond our scope to explore that in this paper.
> > > > >
> > > > > In summary, at within-clip level, it is non-trivial to directly adopt any attention mechanisms designed for video classification or image recognition in order to improve the within-clip tracking ability. We carefully design our axial-trajectory and show the great potential of it in video segmentation as suggested by the table below. More details can be found in Tab. 5 of appendix, where we carefully ablated the within-clip tracking module.
> > > > >
> > > > > model | VPQ | GFlops |
> > > > > |:---:|:---:|:---:|
> > > > > Video-kMaX + MSDeformAttn | 44.5 | 432 |
> > > > > Video-kMaX + MSDeformAttn + trajectory attn * 2 | 45.3 | 494 |
> > > > > Video-kMaX + MSDeformAttn + axial-trajectory attn * 2 | 45.4 | 458 |
> > > > > Video-kMaX + MSDeformAttn + axial-trajectory attn * 4 | 46.1 | 481 |
> > > > >
> > > > > [1] Bertasius et al. Is Space-Time Attention All You Need for Video Understanding
> > > > >
> > > > > [2] Ramachandran et al. Stand-Alone Self-Attention in Vision Models
> > > > >
> > > > > [3] Hassani et al. Neighborhood Attention Transformer
> > > > >
> > > > > [4] Beltagy et al. Longformer: The Long-Document Transformer
> > > > >
> > > > > [5] Xie et al. SegFormer: Simple and Efficient Design for Semantic Segmentation with Transformers
> > > > >
> > > > > [6] Pan et al. Slide-Transformer: Hierarchical Vision Transformer with Local Self-Attention

---

> > > > > ### Author Response · Authors · 2023-11-20
> > > > > **Invitation to articulate your remaining concerns in detail**
> > > > >
> > > > > With the limited remaining rebuttal discussion period, we kindly request Reviewer TBSQ for further clarification to ensure clarity.
> > > > >
> > > > > In our previous response addressing the concern regarding your "*other quadratic computation issues of self-attention* raised by Reviewer tzju", we aim to understand more precisely the ***specific*** "other self-attention" method that might be absent from our discussion or comparison. We earnestly invite you to articulate your remaining concerns in detail, especially if any aspect of our discussion or comparison lacks clarity. We are fully committed to providing a more comprehensive and detailed explanation, if necessary.
> > > > >
> > > > > We genuinely appreciate your time and invaluable feedback throughout this review process.

---

### Author Response · Authors · 2023-11-15
**Common Comments (1/2)**

We appreciate all reviewers for their valuable suggestions, and we address the common concerns as follows. For the remaining concerns, please see the individual post for each reviewer. We kindly point out that we have provided additional visualizations of more axial-trajectory attention maps, failure cases, detailed figures of axial-trajectory attention and Temporal-ASPP as requested by the reviewers in the revised version. Please download it and check ***Section F in the Appendix*** for all the materials.

> C1: From Reviewers TBSQ, tzTu, tzju, Novelty/Contribution

We emphasize that our within-clip tracking module and cross-clip tracking module designs are not a ***trivial*** adaptation from trajectory attention or ***naive*** combination of existing methods. Most importantly, we ***never*** claim trajectory attention as one of our main contributions. We properly phrased our contributions in the paper.

Specifically, our main contribution and novelty lies in that we are the first to exploit trajectory-based temporal self-attention at ***dense pixel prediction tasks*** by extending trajectory attention to ***axial-trajectory*** attention.
As discussed in related work, there have been many works exploring different temporal self-attention mechanisms in **video classification**, but very limited work has been done at **dense video pixel-level prediction** tasks due to the high resolution input in segmentation and intolerable complexity. Most video segmentation works bypass this challenge by exploiting frame-level segmenters and conducting inference in an online manner, which may achieve sub-optimal results. By contrast, our axial-trajectory design effectively brings two advantages:

1. It significantly reduces the computation complexity of trajectory attention, which makes it possible to apply it to the high resolution feature maps. This is also supported by the results in Tab. 5 of appendix, where trajectory attention can only be applied for at most **two** blocks, while axial-trajectory can be applied for **eight** blocks within a single V100 GPU; otherwise, it raises CUDA out-of-memory issues.
2. The introduction of axial-trajectory attention effectively benefits the model to capture the within-clip temporal interactions, thus boosting the performance in both VPS and VIS by a non-trivial margin.

Besides, for our cross-clip tracking module, we also ***do not*** claim ASPP as our novelty. Our main contributions at this stage are twofold:

1. To our knowledge, we are the first to extend the ASPP to the temporal domain and operate on object queries in order to capture the local temporal connections motivated by the design of ASPP, forming the proposed Temporal-ASPP.
2. The cross-clip tracking module ***seamlessly*** integrates trajectory attention and Temporal-ASPP into one module, which effectively reasons for both the global and local cross-clip connections, respectively.

We would like to raise a special focus on the comparison to VITA as suggested by Reviewer tzju. Our cross-clip tracking module exploits a completely different idea for tracking at the video-level. Specifically, VITA introduces an ***additional*** set of video object queries to decode the information out from frame object queries while MaXTron only manipulates and refines the original clip queries. As a result, MaXTron enjoys a much simpler design and achieves better performance. Please refer to Q2.2 in response to Reviewer tzju for the detailed comparison.

In summary, our within-clip tracking module and cross-clip tracking module are both well-motivated and well-designed in order to tackle the key challenges in the video segmentation area, which provide significant performance improvement compared to prior arts. Besides, as plug-in modules, they can be easily integrated into any off-the-shelf clip-level segmenter to boost performance (as demonstrated by our experiments). Lastly, the two tracking modules are small in terms of parameters. The within-clip cross tracking module simply contains 9.1M parameters, while the cross-clip tracking module contains only 7.7M parameters.

> C2: From reviewers TBSQ, tzTu, Failure cases

We thank the reviewer for the suggestions. We have provided failure cases and deeper analysis in the revised version (please download it and check **Fig. 13 and Fig. 14** in the appendix). Here we simply summarize the three common error patterns that we have observed:

1. If there are heavy occlusions caused by many close-by instances, ID switching problem still bothers MaXTron. (i.e., MaXTron assigns inconsistent ID to the same instance in a video as time goes on.)
2. If the motion movement is too large across frames, MaXTron might fail to accurately capture the boundary of the moving object.
3. If extreme illumination (e.g., caused by strong lighting) exists in certain regions, MaXTron might fail to detect and consistently segment out the objects.

---

> ### Author Response · Authors · 2023-11-15
> **Common Comments (2/2)**
>
> > C3: From reviewers LGcv, tzju, Training cost and inference speed
>
> We thank the reviewers for the questions. Since most prior works in this literature do not report these numbers, we can only provide the comparison between MaXTron and our baseline off-the-shelf clip-level segmenters here in terms of training GPU days, model parameters, GFlops and FPS.
>
> We first compare the effect of deploying the proposed *within-clip tracking module* to the baseline Video-kMaX as follows:
>
> model | backbone | input size | GPU days (A100) | params | GFlops | FPS (A100) | VPQ |
> |:---------:|:----------:|:----------:|:---------:|:----------:|:----------:|:----------:|:----------:|
> Video-kMaX     |    ResNet-50    |    $2 \times 769 \times 1345$    |    6.9    |    72.4M    |    354    |    14.3    |    42.7    |
> Video-kMaX + MSDeformAttn     |    ResNet-50    |    $2 \times 769 \times 1345$    |    7.8    |    75.7M    |    432    |    12.5    |    44.5    |
> Video-kMaX + Axial-Trajectory Attn     |    ResNet-50    |    $2 \times 769 \times 1345$    |    7.9    |    80.0M    |    443    |    11.7    |    44.9   |
> MaXTron w/ Video-kMaX (online) |   ResNet-50    |    $2 \times 769 \times 1345$   |    9.5    |    81.5M    |    481    |   10.5    |   46.1   |
>
> As can be observed from the table, the proposed within-clip tracking module contains very few parameters (marginal 9.1M). It brings reasonable increase in training time and GFlops, since it is compute-intensive to conduct attention-based operation on the high-resolution dense pixel feature maps. We note that MSDeformAttn and Axial-Trajectory Attn brings roughly the same amount of computational cost.
>
> Now, we carefully compare the effect of deploying the proposed *cross-clip tracking module* to Video-kMaX. Specifically, we compare MaXTron w/ Video-kMaX (denoted as Ours) vs. VITA w/ Video-kMaX (denoted as VITA) in the following table:
>
> cross-clip tracking design | backbone | input size | GPU days (A100) | params | GFlops (cross-clip tracking module only) | VPQ |
> |:---------:|:----------:|:----------:|:----------:|:----------:|:----------:|:----------:|
> Ours     |    ResNet-50    |    $24 \times 769 \times 1345$    |    3.5    |    7.7M    |    32    |    46.7    |
> VITA     |    ResNet-50    |    $24 \times 769 \times 1345$    |    5.2    |    12.9M    |    47    |    46.3   |
>
> As shown in the table, our cross-clip tracking module is ***more lightweight*** compared to VITA and achieves ***better*** performance, which is brought by the ***much simpler*** design of our cross-clip tracking module. We refer to Q.2.2 to Reviewer tzju for the detailed comparison. FPS is not reported here since offline inference is done on CPU (following the literature) due to the high resolution ($769 \times 1345$) and long video length (24 frames in the table).

---

### Author Response · Authors · 2023-11-18
**Invitation to Engage in Author-Reviewer Discussion for MaXTron**

Dear Reviewers and AC,

We extend our sincere gratitude for the insightful and constructive feedback you have provided on our paper. Your inputs are immensely valuable to us. As the author-reviewer discussion phase has commenced for a while, we are eager to engage with you and address any queries or concerns you may have regarding our responses or the paper itself. We understand that time is of the essence and would greatly appreciate your involvement in this crucial phase of the review process.

Your perspectives and suggestions are pivotal in enhancing the quality and impact of our work. Should you require further elucidation or additional information, we stand ready and enthusiastic to facilitate a productive dialogue.

Thank you once again for your time and consideration.

Warm regards,

Authors of MaXTron

---

### Author Response · Authors · 2023-11-21
**Continued Engagement in the Ongoing Discussion**

Dear Reviewers and AC,

We wholeheartedly invite and encourage further discussion. We kindly request your continued engagement in the ongoing dialogue to ensure a comprehensive and well-rounded review process. Your time and expertise are sincerely appreciated, and we eagerly await your continued input.

Best regards,

MaXTron Authors

---

### Author Response · Authors · 2023-11-23
**Appreciation for Your Dedication and Valuable Insights**

Dear Reviewers, Area Chair, and Senior Area Chair,

As the author-reviewer discussion is drawing to a close, we extend our heartfelt appreciation for your dedicated time and invaluable insights shared throughout the review process.

In our provided rebuttal, we diligently strived to comprehensively address all concerns raised by the reviewers. Regrettably, despite our efforts, Reviewer TBSQ and Reviewer tzju have not engaged further with our latest rebuttal or provided additional clarification for their reviews. It is important to highlight our earnest endeavor in presenting novel designs (within-clip and cross-clip tracking modules) aimed at enhancing existing clip-level segmenters, showcasing state-of-the-art performance.

We kindly request the AC and SAC to consider our extensive efforts and the substantial advancements demonstrated in our work while deliberating on the final decision.

Sincerely,

MaXTron Authors

---

### Meta-Review · Area_Chair_F3fT · 2023-12-06

**Metareview:**

The paper presents MaXTron for video panoptic segmentation.  It employs trajectory attention by leveraging within clips and between clips tracking modules. As the weakness outweighs the merits, given the current status, AC recommends rejection.

Strengths:
1. Paper is well-organized and easy to read
2. The experiments demonstrated an improvement in the performance.

Weakness:
1. The model design is incremental and the novelty is limited, as pointed out by all 4 reviewers. Specifically, the method was incrementally built upon the existing method. There are similar ideas in the literature attempting to reduce the number of tokens with attention mechanisms.
2. The design motivation is unclear. The initial design was to save computing costs; however, relevant compute resource analysis was not provided. 2 reviewers pointed out this issue.
3. minors: clarity and figure designs

**Justification For Why Not Higher Score:**

see weakness

**Justification For Why Not Lower Score:**

N/A

---

### Decision · Program_Chairs · 2024-01-16

Reject